# Nash-GBML: Nash Gradient-Based Meta-Learning

## Abstract

Meta-learning has been proposed to address fast adaptation to unseen tasks with little data. Traditional meta-learning is modeled as the Single-Leader Multi-Follower game consisting of inner and outer-level problems to minimize average or worst-case task loss. Because they assume all sampled tasks are independent, it reduces the flexibility of modeling complex interaction among tasks. Thus, we formulate meta-learning as a Single-Leader Multi-Follower game by considering the interaction among tasks at the inner level. We propose the Nash-GBML incorporating a penalty term into the task loss function to model the interaction among task-specific parameters. We discuss the iteration complexity and convergence of the Nash-GBML algorithm. To validate our Nash-GBML algorithm, we introduce two penalty terms, which are designed to reduce the average and worst-case task loss. We empirically show that the Nash-GBML with the proposed penalty terms outperforms traditional GBML for supervised learning experiments.

## 1 Introduction

In machine learning, fast adaptation to unseen tasks with little data remains challenging. Meta-learning, also known as learning-to-learn (Baxter, 1998), has been proposed to address this issue. According to the framework proposed by Baxter (1998), meta-learning aims to minimize the expected loss on new tasks by leveraging information learned from similar tasks. One approach for meta-learning is gradient-based meta-learning (GBML) (Finn et al., 2017).

In GBML, the parameter for a new task is updated from the meta-parameter with only a few stochastic gradient descent (SGD) steps. Stemming from the Model-Agnostic Meta-Learning (MAML) (Finn et al., 2017), GBML has developed in diverse directions. First-order MAML (FOMAML) (Nichol et al., 2018) learns faster than MAML by ignoring second-order derivatives. Reptile (Nichol et al., 2018) also learns faster by updating the meta-parameter using the expected gradient of meta-training tasks. Meta-SGD (Li et al., 2017) learns more efficiently compared to MAML by learning not just the meta-parameter but also the update direction and learning rate. CAVIA (Zintgraf et al., 2019) is less prone to overfitting by updating context parameters at the inner level, instead of the entire network. Task-robust MAML (TR-MAML) (Collins et al., 2020) is the task-robust meta-learning by minimizing the worst-case task loss, instead of the average task loss at the outer level. These studies first update task-specific parameters independently during meta-training to solve the inner level problem. Then, they update the meta-parameter during meta-testing to solve the outer level problem.

Typical meta-learning optimizes the meta-parameter by minimizing the average task loss or the worst-case task loss, assuming that all sampled tasks are independent. The assumption can greatly simplify the optimization of the meta-parameter; however, it reduces the flexibility of modeling complex interaction among tasks. We hypothesize that considering the interaction among tasks will affect the updating process of the meta-parameter and the overall performance. Based on this hypothesis, we propose a new meta-learning framework that considers the interaction among meta-training tasks. Here, the interaction refers to each task-specific parameter influencing the optimization of other task-specific parameters, akin to how a single decision maker's decision affects other decision makers' utility in game theory. Thus, we formulate meta-learning as a Single-Leader Multi-Follower (SLMF) game (Xi et al., 2022) where the optimization process of task-specific pa-

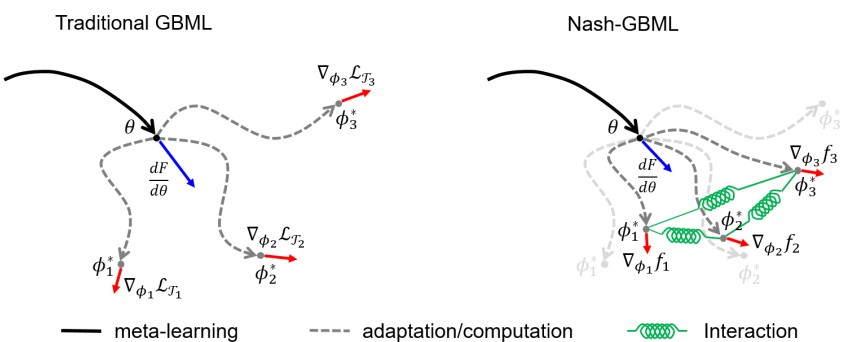

Figure 1: GBML algorithms compute the gradient of the meta-loss function $\frac{dF}{d\theta}$ by using $\nabla_{\phi_i}\mathcal{L}_{\mathcal{T}_i}$. To consider the interaction among tasks, Nash-GBML transforms the task loss function $\mathcal{L}_{\mathcal{T}_i}$ of GBML algorithms into $f_i$ by incorporating terms affected by the joint task-specific parameter. Nash-GBML algorithm computes a Nash equilibrium of the inner level problem and computes the meta-gradient $\frac{dF}{d\theta}$ by using $\nabla_{\phi_i}f_i$.

rameters is the decision-making process of multiple followers, and the optimization process of the meta-parameter is the decision-making process of the leader.

As GBML has been proposed as an effective algorithm to solve the meta-learning problem, we first propose Nash-GBML as an algorithm to address meta-learning problems modeled as the SLMF game. As shown in Figure 1, while GBML assumes that tasks are independent, Nash-GBML considers the interactions among tasks. Thus, solving meta-learning modeled as the SLMF game is equivalent to finding a general Stackelberg equilibrium among the leader and followers. Nash-GBML consists of two levels of equilibrium-finding problems: (1) finding a Nash equilibrium among followers (task-specific parameters) and (2) finding a general Stackelberg equilibrium of the leader (meta-parameter) given a Nash equilibrium among the followers. Because Nash-GBML can consider various forms of interactions among tasks, it is more expressive and capable of capturing complex dependencies among tasks. Next, we discuss the iteration complexity and convergence of the Nash-GBML algorithm.

The interaction among tasks can be modeled in various ways, reflecting the characteristics of the domain. In this study, we introduce two penalty terms applied to the task loss function to model the interaction among the task-specific parameters in a way that is universally applicable across various domains. Because all task-specific parameters should be close to the meta-parameter to ensure they can be obtained through a few gradient update steps, we first design a penalty term to regularize the task-specific parameters to be close to each other. Then, to improve the worst-case performance across tasks, we define another penalty term to apply a stronger regularization to the task-specific parameter that are farther from the meta-parameter. We evaluate the effectiveness of the proposed meta-learning framework with the designed penalty terms and the Nash-GBML algorithm using three benchmark problems.

## 2 PRELIMINARIES

### 2.1 GAME THEORY

Game theory is the discipline that models problems where multiple decision-makers aim to optimize their objectives. Game consists of players who make decisions, their feasible regions (or strategies), and their objective functions (or utilities). In this study, we introduced Nash game (Nash, 1950; 1951) and the Single-Leader Multi-Follower (SLMF) game (Song, 1992; Xi et al., 2022; Jo et al., 2023) to model the meta-learning using game-theoretic concept. First, we discuss the $N$ players' Nash game in which $N$ players make decisions simultaneously.

**Definition 2.1** *Let $G = \left\langle \mathbf{P}, (u_i)_{i\in\mathbf{P}}, (\Omega_i)_{i\in\mathbf{P}} \right\rangle$ be a $N$ players' Nash game. Then, player $i \in \mathbf{P}$ makes their decisions to maximize the utility function $u_i$:*

$$\max_{\mathbf{x}_i \in \Omega_i} u_i\left(\mathbf{x}_i, \mathbf{x}_{-i}\right) \tag{1}$$

*where $\mathbf{P} = \{1, \cdots, N\}$ is the set of players, $u_i$ is the utility function of the player $i$, $\mathbf{x}_i$ is the player $i$'s decision belonging to the strategy set $\Omega_i$, and $\mathbf{x}_{-i} = (\mathbf{x}_1, \cdots, \mathbf{x}_{i-1}, \mathbf{x}_{i+1}, \cdots, \mathbf{x}_N)$ is the player's joint decision except player $i$. Then, we refer to $\mathbf{x}^* \in \prod_{i\in\mathbf{P}} \Omega_i$ as a Nash equilibrium of the $N$ player's Nash game $G$ if it satisfies the following equation.*

$$\mathbf{x}_i^* = \arg\max_{\mathbf{x}_i \in \Omega_i} u_i\left(\mathbf{x}_i, \mathbf{x}_{-i}^*\right), \forall i \in \mathbf{P} \tag{2}$$

When the player $i$'s strategy set $\Omega_i$ is dependent to other players' decisions $\mathbf{x}_{-i}$, we refer to $G$ as a generalized Nash game (Facchinei & Kanzow, 2010), and $\mathbf{x}^* \in \prod_{i\in\mathbf{P}} \Omega_i\left(\mathbf{x}_{-i}^*\right)$ as a generalized Nash equilibrium.

Next, we discuss the SLMF game where a leader makes decisions first, and then $N$ followers make their decisions simultaneously after observing the leader's decision.

**Definition 2.2** *Let $\Gamma = \left\langle \{1\}, \mathbf{F}, u^{\mathrm{L}}, (u_i)_{i\in\mathbf{F}}, \Omega^{\mathrm{L}}, (\Omega_i)_{i\in\mathbf{F}} \right\rangle$ be the SLMF game. First, a leader makes their decisions to maximize the utility function $u^{\mathrm{L}}$:*

$$\max_{\mathbf{y}\in\Omega^{\mathrm{L}}} u^{\mathrm{L}}\left(\mathbf{y}, (\mathbf{x}_i)_{i\in\mathbf{F}}\right) \tag{3}$$

*where $\mathbf{F} = \{1, \cdots, N\}$ is the set of followers, $u^{\mathrm{L}}$ is the utility function of a leader, and $\mathbf{y}$ is the leader's decision belonging to the strategy set $\Omega^{\mathrm{L}}$. Then, $N$ followers make their decisions simultaneously to maximize the utility function:*

$$\max_{\mathbf{x}_i\in\Omega_i(\mathbf{y})} u_i\left(\mathbf{y}, \mathbf{x}_i, \mathbf{x}_{-i}\right) \tag{4}$$

*where $u_i$ is the utility function of follower $i$, and $\mathbf{x}_i$ is the follower $i$'s decision belonging to the strategy set $\Omega_i$. Then, we refer to $\left(\mathbf{y}^*, (\mathbf{x}_i^*)_{i\in\mathbf{F}}\right) \in \Omega^{\mathrm{L}} \times \prod_{i\in\mathbf{F}} \Omega_i\left(\mathbf{y}^*\right)$ of $\Gamma$ as a general Stackelberg equilibrium if it satisfies the following equation.*

$$\sup_{\mathbf{x}\in\mathbf{S}(\mathbf{y}^*)} u^{\mathrm{L}}\left(\mathbf{y}^*, \mathbf{x}\right) \geq \inf_{\mathbf{x}\in\mathbf{S}(\mathbf{y})} u^{\mathrm{L}}\left(\mathbf{y}, \mathbf{x}\right), \forall \mathbf{y} \in \Omega^{\mathrm{L}} \tag{5}$$

where $\mathbf{x} = (\mathbf{x}_i)_{i\in\mathbf{F}}$ is the followers' joint decision, and $\mathbf{S}(\mathbf{y})$ is the set of a (generalized) Nash equilibrium of the $N$ followers' (generalized) Nash game given leader's decision $\mathbf{y}$, $G(\mathbf{y}) = \left\langle \mathbf{F}, (u_i)_{i\in\mathbf{F}}, (\Omega_i)_{i\in\mathbf{F}} \right\rangle$.

## 2.2 GAME THEORETIC INTERPRETATION OF GRADIENT-BASED META-LEARNING

Traditional meta-learning is generally modeled as the Single-Leader Multi-Follower (SLMF) game $\Gamma = \left\langle \{1\}, [N], F, (\mathcal{L}_{\mathcal{T}_i})_{i\in[N]}, \mathbb{R}^d, \mathbb{R}^d \right\rangle$ where $N$ is the number of task, $F$ is the meta-loss function, and $\mathcal{L}_{\mathcal{T}_i}$ is the task $i$'s loss function.

$$\text{Leader: } \min_{\theta\in\mathbb{R}^d} F(\theta, \phi) := \frac{1}{N}\sum_{i=1}^{N} \mathcal{L}\left(\theta, \phi_i; \mathcal{D}_i^{\text{test}}\right) \tag{6}$$

$$\text{Follower: } \min_{\phi_i\in\mathbb{R}^d} \mathcal{L}\left(\theta, \phi_i; \mathcal{D}_i^{\text{tr}}\right), \forall i \in [N] \tag{7}$$

where $\phi = (\phi_i)_{i\in[N]}$ is the joint task-specific parameter, $N$ is the number of tasks, $\mathcal{D}_i^{\text{test}}$ is the task $i$'s meta-testing data, $\mathcal{D}_i^{\text{tr}}$ is the task $i$'s meta-training data, $\mathcal{L}$ is the task loss function, and $d$ is the dimension of the meta-parameter.

To compute a general Stackelberg equilibrium of the SLMF game described in equations (6) and (7), MAML with a one-step gradient update first approximates the optimal task-specific parameter of the inner level problem as the one SGD update from the meta-parameter. Then, it computes the

gradient of the meta-loss $\frac{dF}{d\theta}$ of the outer level problem by approximating the meta-loss function $F$ as the average of meta-training task losses.

$$\theta^* := \arg\min_{\theta \in \mathbb{R}^d} F\left(\theta, \phi^*\left(\theta\right)\right) \approx \frac{1}{B} \sum_{i=1}^{B} \mathcal{L}\left(\phi_i^*\left(\theta\right); \mathcal{D}_i^{\text{test}}\right) \tag{8}$$

$$\phi_i^*\left(\theta\right) := \arg\min_{\phi_i \in \mathbb{R}^d} \mathcal{L}\left(\theta, \phi_i; \mathcal{D}_i^{\text{tr}}\right) \approx \theta - \alpha \frac{d}{d\phi_i} \mathcal{L}\left(\phi_i, \mathcal{D}_i^{\text{tr}}\right)|_{\phi_i=\theta}, \forall i \in [N] \tag{9}$$

where $\alpha$ is the inner learning rate, and $B$ is the batch size.

The game theoretic interpretation for the other famous GBML algorithms such as Meta-SGD (Li et al., 2017), iMAML (Rajeswaran et al., 2019), CAVIA (Zintgraf et al., 2019), and TR-MAML (Collins et al., 2020) are discussed in Appendix A.

## 3 NASH GRADIENT-BASED META-LEARNING

### 3.1 PROBLEM FORMULATION

Traditional meta-learning independently optimizes each task-specific parameter at the inner level. We extend the meta-learning framework by considering the interaction among meta-training tasks at the inner level. In this study, we combine a penalty term $p\left(\theta, \phi_i, \phi_{-i}\right)$ into the task loss $\mathcal{L}_{\mathcal{T}_i}\left(\theta, \phi_i\right)$ for each task $i$ to account for the influence of the other task-specific parameters $\phi_{-i}$ during meta-training. Thus, the new meta-training framework can be formulated as the SLMF game $\Gamma = \left\langle \{1\}, [N], F, (f_i)_{i \in [N]}, \Omega^{\text{L}}, (\Omega_i)_{i \in [N]} \right\rangle$ where $N$ is the number of tasks, $f_i$ is the inner level objective function, and $\phi = (\phi_i)_{i \in [N]}$ is the joint task-specific parameter.

$$\text{Leader:} \quad \min_{\theta \in \Omega^{\text{L}}} F\left(\theta, \phi\right) \tag{10}$$

$$\text{Follower:} \quad \min_{\phi_i \in \Omega_i} f_i\left(\theta, \phi\right), \forall i \in [N] \tag{11}$$

where the task $i$'s objective function for the inner level problem $f_i$ is defined as the sum of task loss $\mathcal{L}_{\mathcal{T}_i}$ and the penalty term $p$ to be affected by other task-specific parameters $\phi_{-i}$.

$$f_i\left(\theta, \phi\right) = \mathcal{L}\left(\theta, \phi_i; \mathcal{D}_i^{\text{tr}}\right) + p\left(\theta, \phi_i, \phi_{-i}; \mathcal{D}_i^{\text{tr}}\right) \tag{12}$$

We account for the interactions among tasks at the inner level through a penalty term. We discuss the penalty terms in detail in Section 3.3. The inner level problem described in equation (11) is modeled as the $N$ players' Nash game $G\left(\theta\right) = \left\langle [N], (f_i)_{i \in [N]}, (\Omega_i)_{i \in [N]} \right\rangle$.

### 3.2 ALGORITHM

In Nash-GBML, we approximately sample $B$ tasks to compute the general Stackelberg equilibrium of the SLMF game $\Gamma$ described in equations (10) and (11). That is, we compute a Nash equilibrium of the $B$ players' Nash game $\hat{G}\left(\theta\right) = \left\langle [B], (f_i)_{i \in [B]}, (\Omega_i)_{i \in [B]} \right\rangle$ at the inner level to approximate a Nash equilibrium of $G\left(\theta\right) = \left\langle [N], (f_i)_{i \in [N]}, (\Omega_i)_{i \in [N]} \right\rangle$, then update the meta-parameter by approximating the meta-loss $F$ as the average or the worst-case task loss for meta-training tasks.

**Inner level problem.** During meta-training, Nash-GBML algorithm first computes a Nash equilibrium $\phi^{(t)}$ of the Nash game $\hat{G}\left(\theta^{(t)}\right) = \left\langle [B], (f_i)_{i \in [B]}, (\Omega_i)_{i \in [B]} \right\rangle$ using Algorithm 2. Algorithm 2 is based on well-known algorithms for computing Nash equilibrium such as the proximal-decomposition algorithm (Scutari et al., 2012; Atzeni et al., 2013), regularized NI-function type method (Facchinei & Kanzow, 2010; Jo & Park, 2020), or the projected reflected gradient descent (PRGD) method (Malitsky, 2015). Nash-GBML algorithm approximates a Nash equilibrium $\phi^{(t)}$ by $n$-step gradient update.

**Outer level problem.** During meta-testing, Nash-GBML algorithm updates the meta-parameter $\theta^{(t)}$ using $\frac{d\phi^{(t)}}{d\theta}$, which is computed through back-propagation of $\phi^{(t)}$ in Algorithm 1. Therefore, the solution computed by Nash-GBML algorithm is approximately the general Stackelberg equilibrium of the SLMF game $\Gamma = \left\langle \{1\}, [N], F, (f_i)_{i \in [N]}, \Omega^{\mathrm{L}}, (\Omega_i)_{i \in [N]} \right\rangle$. Nash-GBML for MAML, Meta-SGD, and CAVIA approximates the meta-loss function as $F(\theta, \phi) \approx \frac{1}{B} \sum_{i=1}^{B} \mathcal{L}(\theta, \phi_i; \mathcal{D}_i^{\mathrm{test}})$, and Nash-GBML for TR-MAML approximates it as $F(\theta, \phi) \approx \max_{i \in [B]} \mathcal{L}(\theta, \phi_i; \mathcal{D}_i^{\mathrm{test}})$.

---

**Algorithm 1** Nash-GBML

---

**Require :** Distribution over tasks $p(\mathcal{T})$, outer learning rate $\beta$
Randomly initialize meta-parameters $\theta^{(0)}$, $t = 0$
**while** not done **do**
    Sample batch of tasks $\{\mathcal{T}_i\}_{i=1}^{B} \sim p(\mathcal{T})$
    Compute joint task-specific parameter $\phi^{(t)} = $ **Nash-Equilibrium** $\left( \{\mathcal{T}_i\}_{i=1}^{B}, \theta^{(t)} \right)$
    Update $\theta^{(t+1)} \leftarrow \theta^{(t)} - \beta \nabla_{\theta \in \Omega^{\mathrm{L}}} F(\theta^{(t)}, \phi^{(t)})$
    $t \leftarrow t + 1$
**end while**

---

**Algorithm 2** Nash-Equilibrium

---

**Require :** Inner learning rate $\alpha$, number of gradient updates $n$
**Input :** Batch of tasks $\{\mathcal{T}_i\}_{i=1}^{B}$, meta-parameter $\theta$
Initialize task-specific parameter $\phi_i = \theta, \forall i \in [B]$
**for** $k \in \{0, 1, \cdots, n-1\}$ **do**
    **for all** $i \in [B]$ **do**
        $\phi_i \leftarrow \phi_i - \alpha \nabla_{\phi_i \in \Omega_i} f_i(\theta, \phi)$
    **end for**
**end for**
**Return :** $\phi$

---

### 3.3 PENALTY TERMS

Because Nash-GBML is the framework that incorporates a penalty term $p$ into the task loss $\mathcal{L}$ for the inner level problem of traditional GBML, the interaction among tasks is determined by the structure of the penalty term. While the structure of the penalty term varies depending on the domain, we introduce the following two penalty terms, which are universally applicable across various domains as a guideline. Note that the following penalty terms are simply examples, and you can design your own penalty term depending on the domain characteristics.

Because meta-learning is for similar tasks sampled from the same task distribution, each task-specific parameter $\phi_i$ should be close to each other for effective few-shot adaptation to fit new tasks. Thus, we expect that adjusting the meta-parameter and each task-specific parameter to be closer in the inner level problem will minimize the meta-loss function $F$ in the outer level problem. Moreover, it serves as a clamp function for the gradient of the meta-loss function, allowing the algorithm to converge more stably. Based on this hypothesis, we design two penalty terms to penalize the distance between the meta-parameter $\theta$ and task-specific parameters $\phi_i$.

**Centroid penalty term.** The centroid penalty term $p_C(\theta, \phi_i, \phi_{-i}; w)$ with a weight $w$ is designed to penalize the distance between the meta-parameter and the center of task-specific parameters $\frac{1}{B} \sum_{k=1}^{B} \phi_k$. Because it is a penalty that shifts the meta-parameter toward the center of the task-specific parameters, it sometimes negatively impact the worst-case task loss, but it reduces the average task

Table 1: Complexity for the gradient-based meta-learning

| GBML Algorithm | Iteration complexity | Memory |
|---|---|---|
| MAML (GD, full back-prop) | $\kappa \log (D/\delta)$ | $\text{Mem} (\nabla \mathcal{L}) \kappa \log (D/\delta)$ |
| MAML (Nesterov's AGD, full back-prop) | $\sqrt{\kappa} \log (D/\delta)$ | $\text{Mem} (\nabla \mathcal{L}) \sqrt{\kappa} \log (D/\delta)$ |
| implicit MAML (Nesterov's AGD) | $\sqrt{\kappa} \log (D/\delta)$ | $\text{Mem} (\nabla \mathcal{L})$ |
| Nash-GBML (PRGD, full back-prop) | $\kappa \log (D/\delta)$ | $\text{Mem} (\nabla \mathcal{L}) \kappa \log (D/\delta)$ |

loss. The centroid penalty term is defined as follows:

$$p_C (\theta, \phi_i, \phi_{-i}; w) = w \left( \frac{B}{\alpha N} \right)^2 \left\| \theta - \frac{1}{B} \sum_{k \neq i} \phi_k - \frac{1}{B} \phi_i \right\|_2^2 \tag{13}$$

where $\alpha$ is the inner learning rate, $N$ is the number of task for domain, and $B$ is the batch size. $\left( \frac{B}{\alpha N} \right)^2$ is the proportional constant that ensures robustness to the algorithm's hyper-parameters. The detailed explanation for the proportional constant is provided in Appendix E.

**Robust penalty term.** Next, we introduce the robust penalty term $p_R (\theta, \phi_i, \phi_{-i}; w, r)$ with weight $w$ and robustness constant $r$. The robust penalty term $p_R$ is designed to impose a stronger penalty on tasks that are farther from the meta-parameter $\theta$ by penalizing the distance between the meta-parameter and a task-specific parameter $\|\theta - \phi_i\|$ relative to the sum of its distance $\sum_{k=1}^{B} \|\theta - \phi_k\|$. That is, we define the robust penalty term as follows:

$$p_R (\theta, \phi_i, \phi_{-i}; w, r) = w \frac{B \|\theta - \phi_i\|_2^{2r}}{C + \sum_{k=1}^{B} \|\theta - \phi_k\|_2^{2r}} \tag{14}$$

where $C$ is a very small constant to prevent the denominator from becoming zero. As the robustness constant $r$ increases, we impose a stronger penalty on tasks farther from the meta-parameter.

To verify whether the proposed penalty terms work well to achieve the designed purposes, we evaluate the Nash-GBML, which combines traditional GBML with the proposed penalty terms through few-shot supervised learning experiments in Section 4.

### 3.4 ANALYSIS

First, we discuss the iteration complexity of the Nash-GBML algorithm by Theorem 3.1 when we use the projected-reflected gradient-descent (PRGD) method (Malitsky, 2015) to compute a Nash equilibrium. Table 1 summarizes the iteration complexity to compute $\frac{d\phi^*}{d\theta}$ of traditional GBML (Finn et al., 2017; Rajeswaran et al., 2019) and Nash-GBML with PRGD method. Note that the iteration complexity to compute meta-gradient for Nash-GBML is equivalent to traditional GBML algorithms as $O (\log (D/\delta))$.

**Theorem 3.1** *Let $D$ be the diameter of search space of the joint task-specific parameter $\phi = (\phi_i)_{i \in [B]}$ in the inner level problem (i.e. $\|\phi - \phi^*\| \leq D$). Suppose that the PRGD method is used to compute $\delta$-accurate estimation of the optimal joint task-specific parameter $\hat{\phi} = \left( \hat{\phi}_i \right)_{i \in [B]}$ of a Nash equilibrium, which is the convergent point of the inner level of Nash-GBML algorithm. Under Assumption B.1, Nash-GBML algorithm computes $\hat{\phi}$ with $O (\kappa \log (D/\delta))$ number of iterations, and only $O (\text{Mem} (\nabla \mathcal{L}_i) \kappa \log (D/\delta))$ memory is required where $\kappa$ is the condition number.*

Next, we discuss the convergence of the Nash-GBML algorithm. We discuss the convergence criterion and convergence speed in Theorem 3.2. Then, we prove that the Nash-GBML algorithm always converges to the general Stackelberg equilibrium of the SLMF game regardless of the gradient update order of the task-specific parameters in the inner level, the initial meta-parameter, and the initial task-specific parameters in Theorem 3.3.

Table 2: MSE on 5-shot 3-step sinusoid regression with 95% confidence intervals over 5 random trials. The bold values indicate the best performance metric for each of the GBML algorithms.

| Algorithm | Mean | Worst | Std. Dev. |
|---|---|---|---|
| MAML (Finn et al., 2017) | $0.59 \pm 0.01$ | $3.18 \pm 0.86$ | $0.57 \pm 0.04$ |
| MAML $+p_C$ (0.1) | $\mathbf{0.54 \pm 0.01}$ | $3.00 \pm 0.79$ | $0.55 \pm 0.03$ |
| MAML $+p_R$ (0.01, 2) | $0.56 \pm 0.01$ | $\mathbf{2.81 \pm 0.59}$ | $0.51 \pm 0.02$ |
| Meta-SGD (Li et al., 2017) | $0.19 \pm 0.01$ | $1.48 \pm 0.68$ | $0.18 \pm 0.03$ |
| Meta-SGD $+p_C$ (0.1) | $\mathbf{0.14 \pm 0.00}$ | $1.32 \pm 0.67$ | $0.17 \pm 0.02$ |
| Meta-SGD $+p_R$ (0.00001, 1) | $\mathbf{0.14 \pm 0.00}$ | $\mathbf{1.07 \pm 0.37}$ | $0.16 \pm 0.01$ |
| CAVIA (Zintgraf et al., 2019) | $0.14 \pm 0.01$ | $1.29 \pm 0.62$ | $0.14 \pm 0.02$ |
| CAVIA $+p_C$ (0.001) | $\mathbf{0.13 \pm 0.00}$ | $\mathbf{1.15 \pm 0.59}$ | $0.14 \pm 0.02$ |
| CAVIA $+p_R$ (0.00001, 2) | $\mathbf{0.13 \pm 0.01}$ | $1.18 \pm 0.46$ | $0.15 \pm 0.02$ |
| TR-MAML (Collins et al., 2020) | $0.62 \pm 0.02$ | $2.35 \pm 0.46$ | $0.29 \pm 0.02$ |
| TR-MAML $+p_C$ (0.01) | $\mathbf{0.51 \pm 0.02}$ | $2.33 \pm 1.23$ | $0.28 \pm 0.04$ |
| TR-MAML $+p_R$ (0.01, 1) | $0.52 \pm 0.01$ | $\mathbf{2.20 \pm 1.12}$ | $0.26 \pm 0.04$ |

**Theorem 3.2 (Informal Statement)** *Let $\delta$ and $\bar{\delta}$ be the convergence criterion of the inner level and the outer level, respectively. Under convexity assumption, the Nash-GBML algorithm with outer learning rate $\beta \leq \frac{\bar{\delta}}{\sqrt{4C^2\delta^2 + 4\left(\bar{C}_1 + \frac{C_1 C_2}{\mu_1 + \mu_2}\right)^2}}$ is converged to the general Stackelberg equilibrium of the SLMF game with convergence speed $O\left(\max\left\{k_b^2, k_\sigma^2, k_{\bar{\sigma}}^2\right\}\right)$.*

**Theorem 3.3 (Informal Statement)** *Under the convexity assumption, the Nash-GBML algorithm converges to the same optimal solution regardless of the gradient update order of the task-specific parameters in the inner level, the initial meta-parameter, and the initial task-specific parameters.*

The proof of Theorem 3.1 is provided in Appendix C. The formal statement and the proof of Theorems 3.2 and 3.3 are provided in Appendix D.

## 4 EXPERIMENTS

In this section, we aim to validate the following questions regarding Nash-GBML experimentally: (1) Do the penalty terms work well for its intended purpose? (2) Is Nash-GBML generally applicable to the traditional GBML algorithms such as MAML, Meta-SGD, CAVIA, and TR-MAML? (3) Can we apply Nash-GBML across various domains such as regression and classification?

In this section, we set the same network architecture and same hyper-parameters of both the traditional GBML and Nash-GBML for a fair comparison. The detail explanation for the network architecture and experiment setting are described in Appendix F.

### 4.1 SINUSOID REGRESSION

We evaluate the performance of traditional GBML algorithms (Finn et al., 2017; Li et al., 2017; Zintgraf et al., 2019; Collins et al., 2020) and the Nash-GBML algorithm with two penalty terms $p_C(w)$ and $p_R(w, r)$ in few-shot sinusoid regression problem (Collins et al., 2020). The target is a sine function $y = a \sin(x + b)$ on $x \in [-5, 5]$ with amplitude $a \in [0.1, 5]$ and phase $b \in [0, \pi]$. The amplitude follows the uniform distribution on interval $[0.1, 1.05] \cup [4.95, 5]$ for meta-training, $[0.1, 5]$ for meta-testing. The phase follows the uniform distribution on $[0, \pi]$, and each task consists of $K = 5$ samples where inputs are uniformly sampled from $[-5, 5]$.

We partition the amplitude interval $[0.1, 5]$ into 490 distinct subintervals of length 0.01, and each task is defined as the subintervals. During the meta-testing process, we randomly sample 5000 tasks and computed the test loss for each of the 490 subintervals. Table 2 shows the average MSE, worst MSE, and standard deviation of subintervals with its 95% confidence interval over 5 random trials.

In Table 2, we show that Nash-GBML algorithms consistently outperform the traditional GBML algorithms in both average and worst-case MSE. In Nash-GBML, the weight of the cetnroid penalty

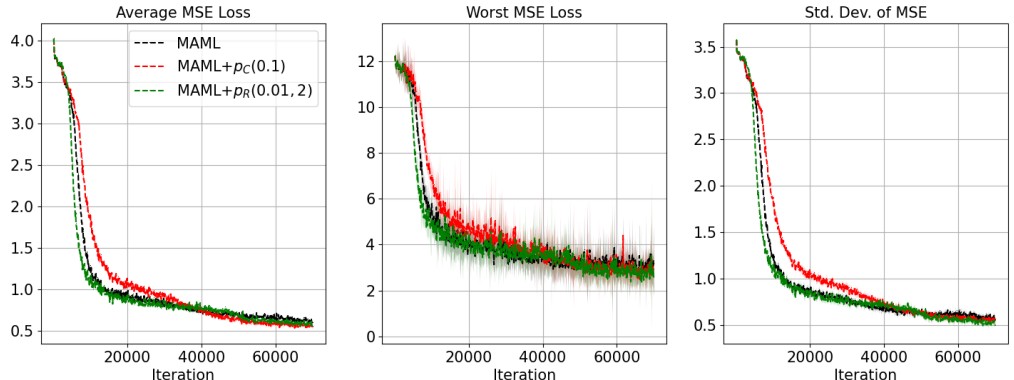

Figure 2: Test MSE statistics with 95% confidence intervals over 5 random trials of MAML and its Nash-GBML. The leftmost plot shows the average MSE loss, the middle plot shows the worst MSE loss, and the rightmost plot shows the standard deviation. The statistics are empirical averages over 5000 samples.

Table 3: MSE on 100-shot 5-step image regression with 95% confidence intervals over 5 random trials. The bold values indicate the best performance metric for each of the GBML algorithms.

| Algorithm | Mean $\left[\times 10^{-2}\right]$ | Worst $\left[\times 10^{-2}\right]$ | Std. Dev. $\left[\times 10^{-2}\right]$ |
|---|---|---|---|
| MAML (Finn et al., 2017) | $1.62 \pm 0.02$ | $6.25 \pm 1.43$ | $0.70 \pm 0.02$ |
| MAML $+p_C$ (0.01) | $\mathbf{1.59 \pm 0.01}$ | $5.94 \pm 0.94$ | $0.68 \pm 0.02$ |
| MAML $+p_R$ (0.0001, 1) | $1.61 \pm 0.01$ | $\mathbf{5.91 \pm 1.26}$ | $0.69 \pm 0.02$ |
| Meta-SGD (Li et al., 2017) | $1.64 \pm 0.02$ | $6.02 \pm 1.28$ | $0.71 \pm 0.01$ |
| Meta-SGD $+p_C$ (0.01) | $\mathbf{1.63 \pm 0.02}$ | $5.72 \pm 0.58$ | $0.70 \pm 0.02$ |
| Meta-SGD $+p_R$ (0.01, 2) | $1.69 \pm 0.02$ | $\mathbf{5.69 \pm 0.54}$ | $0.73 \pm 0.01$ |
| CAVIA (Zintgraf et al., 2019) | $1.85 \pm 0.02$ | $6.53 \pm 0.52$ | $0.80 \pm 0.01$ |
| CAVIA $+p_C$ (0.1) | $1.65 \pm 0.02$ | $6.57 \pm 2.04$ | $0.71 \pm 0.02$ |
| CAVIA $+p_R$ (0.00001, 2) | $\mathbf{1.62 \pm 0.02}$ | $\mathbf{5.96 \pm 0.53}$ | $0.70 \pm 0.01$ |

term, the weight and robustness constant of the robust penalty term are chosen from $10^{-1}$ to $10^{-4}$, $10^{-2}$ to $10^{-5}$, and 1 or 2, respectively. As described in section 3.3, we validate that the centroid penalty term $p_C(w)$ is effective in reducing the average MSE, while the robust penalty term $p_R(w, r)$ is effective in reducing the worst-case MSE in simple regression tasks.

We also evaluate the convergence of Nash-GBML algorithm during training process in Figure 2. Nash-GBML algorithm which combines MAML with two penalty terms consistently outperform MAML after sufficient iterations. In Appendix F, we evaluate the convergence of other Nash-GBML algorithms during training process.

In Appendix F, we compare the traditional GBML and Nash-GBML algorithms depending on the hyper-parameters for the penalty terms, batch size, and step size and verify that Nash-GBML outperforms GBML in most cases. These results show that the penalty terms work well and Nash-GBML is generally applicable to the traditional GBML algorithms on the simple regression task. As described in section 3.3, we also validate that Nash-GBML algorithm converges more stably than the traditional MAML based on the lower MSE standard deviation and the convergence trajectory.

## 4.2 IMAGE COMPLETION

To verify whether Nash-GBML can be applied to challenging regression tasks, we compare it against MAML, Meta-SGD, and CAVIA on the image completion task (Garnelo et al., 2018) using the CelebA domain (Liu et al., 2015). The CelebA contains 162770 training images, 19867 validation images, and 19962 test images. Each task consists of $K = 100$ random pixels per image.

Table 4: 5-way 5-shot classification accuracies on MiniImageNet with $95\%$ confidence intervals over 5 random trials. The bold values indicate the best performance metric for each of the GBML algorithms.

| Algorithm | Mean | Worst | Std. Dev. |
|---|---|---|---|
| MAML (Finn et al., 2017) | $57.20 \pm 0.60\%$ | $29.87 \pm 7.50\%$ | $8.57 \pm 0.74\%$ |
| MAML $+p_C\left(10^4\right)$ | $\mathbf{58.15 \pm 0.43}\%$ | $\mathbf{35.20 \pm 3.14}\%$ | $8.61 \pm 0.45\%$ |
| MAML $+p_R\left(10^{-1}, 1\right)$ | $57.85 \pm 0.29\%$ | $32.80 \pm 4.25\%$ | $8.63 \pm 0.53\%$ |
| Meta-SGD (Li et al., 2017) | $56.96 \pm 0.32\%$ | $30.67 \pm 5.23\%$ | $8.71 \pm 0.27\%$ |
| Meta-SGD $+p_C\left(10^6\right)$ | $58.08 \pm 0.40\%$ | $30.40 \pm 5.07\%$ | $8.69 \pm 0.30\%$ |
| Meta-SGD $+p_R\left(10^{-2}, 1\right)$ | $\mathbf{58.28 \pm 0.33}\%$ | $\mathbf{33.33 \pm 6.61}\%$ | $8.43 \pm 0.84\%$ |
| CAVIA(32) (Zintgraf et al., 2019) | $56.87 \pm 0.99\%$ | $29.33 \pm 4.67\%$ | $8.89 \pm 0.44\%$ |
| CAVIA(32) $+p_C\left(10^3\right)$ | $\mathbf{58.18 \pm 0.45}\%$ | $29.07 \pm 5.33\%$ | $9.21 \pm 0.29\%$ |
| CAVIA(32) $+p_R\left(10^{-1}, 2\right)$ | $57.76 \pm 0.74\%$ | $\mathbf{31.47 \pm 4.85}\%$ | $9.07 \pm 0.50\%$ |
| CAVIA(128) (Zintgraf et al., 2019) | $63.10 \pm 0.76\%$ | $34.93 \pm 4.50\%$ | $8.77 \pm 0.32\%$ |
| CAVIA(128) $+p_C\left(10^4\right)$ | $63.15 \pm 0.48\%$ | $33.87 \pm 7.86\%$ | $8.93 \pm 0.50\%$ |
| CAVIA(128) $+p_R\left(10^{-2}, 1\right)$ | $\mathbf{63.39 \pm 0.50}\%$ | $\mathbf{35.66 \pm 5.73}\%$ | $8.68 \pm 0.67\%$ |

Table 3 shows the MSE statistics for CelebA dataset. In Nash-GBML, the weight of the centroid penalty term, the weight and robustness constant of the robust penalty term are determined from $10^{-1}$ to $10^{-4}$, $10^{-2}$ to $10^{-5}$, and 1 or 2, respectively. Although the batch size($= 25$) is very small relative to the total number of tasks($= 162770$), Nash-GBML outperforms the traditional GBML in both the average and the worst-case MSE in most settings. In particular, we validate the hypothesis that the centroid penalty term reduces the average MSE, and the robust penalty term reduces the worst-case MSE in complex regression tasks, which is described in section 3.3.

### 4.3 CLASSIFICATION

To evaluate Nash-GBML on large-scale classification problem, we compare MAML, Meta-SGD, CAVIA(32), and CAVIA(128) and its Nash-GBML algorithms on the MiniImageNet domain (Ravi & Larochelle, 2017). The MiniImageNet contains 64 training classes, 12 validation classes, and 24 test classes. In $N$-way $K$-shot classification, $N$ classes are randomly chosen, and $K$ samples are randomly chosen from the $N$ classes.

Table 4 shows 5-way 5-shot accuracies for MiniImageNet. In Nash-GBML, we choose the weight and the robustness constant of two penalty terms from $10^1$ to $10^6$, $10^{-1}$ to $10^{-3}$, and 1 or 2, respectively. Although the batch size ($= 2$) is very small relative to the total number of tasks ($= \binom{64}{5}$), Nash-GBML algorithms successfully learn the interaction among tasks and outperform the traditional GBML algorithms in most cases. As described in section 3.3, we also validate that the centroid penalty consistently improves the average accuracy, while the robust penalty consistently improves the worst-case accuracy in complex classification tasks. In conclusion, the Nash-GBML algorithm is generally applicable to the traditional algorithms, and we apply it across various domains.

## 5 CONCLUSION AND FUTURE WORK

We propose the meta-learning framework, which extends traditional meta-learning at the inner level by considering the interaction among tasks. To account for the interaction among tasks at the inner level, we incorporate a penalty term, which is affected by the joint task-specific parameters, into the loss of traditional meta-learning. As a guideline, we introduce two penalty terms, which are designed to reduce the average and worst-case task loss. Then, we experimentally validate that (1) the proposed penalty terms work well to achieve the designed purposes and (2) Nash-GBML is generally applicable to traditional GBML in various domains.

Because Nash-GBML is the framework generally applicable to GBML algorithms, it can be applied to the other GBML algorithms not mentioned in this study. Moreover, because the proposed penalty terms do not reflect the characteristics of the domain, researchers can freely design new penalty

terms according to the target domain. In conclusion, this study proposes a new framework for meta-learning that considers interaction among tasks, and the methodology to support it. We presents a pathway for further expansion whenever a novel GBML algorithms or new penalty terms reflecting domain characteristics are developed.

We are interested in extending the meta-learning framework not only by adding a penalty term to the objective function but also by adding a joint constraint on the task-specific parameters to the strategy set in the inner level problem. We can model it as the generalized SLMF game. In the future, we plan to apply Nash-GBML to other complex regression, classification, and RL domains. We also plan to design a new penalty term that reflects the characteristics of the domain.

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

## A   GAME THEORETIC INTERPRETATION OF GBML

GBML algorithms (Li et al., 2017; Rajeswaran et al., 2019; Zintgraf et al., 2019; Collins et al., 2020) based on MAML (Finn et al., 2017) update the meta-parameter $\theta$ in the outer level and update the task-specific parameter $\phi$ in the inner level. We discuss what problems current GBML algorithms aim to solve. Let $\mathcal{D}_i^{\text{test}}$ be a task $i$'s meta-testing data, $\mathcal{D}_i^{\text{tr}}$ be a task $i$'s meta-training data, $\mathcal{L}$ be a loss function, $\alpha$ be a inner learning rate, $N$ is the number of tasks, $B$ be a batch size, and $d$ be a dimension of meta-parameter.

**Meta-SGD** (Li et al., 2017) learns the inner learning rate in the outer level. Thus, the target problem of Meta-SGD can be modeled as the following Single-Leader Multi-Follower (SLMF) game:

$$\text{Leader:} \quad \min_{\theta \in \mathbb{R}^d, \alpha \in \mathbb{R}^d} F\left(\theta, \alpha, \phi\right) := \frac{1}{N} \sum_{i=1}^{N} \mathcal{L}\left(\theta, \alpha, \phi_i; \mathcal{D}_i^{\text{test}}\right) \tag{15}$$

$$\text{Follower:} \quad \min_{\phi_i \in \mathbb{R}^d} \mathcal{L}\left(\theta, \alpha, \phi_i; \mathcal{D}_i^{\text{tr}}\right) \tag{16}$$

To compute a Stackelberg equilibrium of the SLMF game, Meta-SGD with one-step gradient update first approximates the optimal task-specific parameter $\phi_i^*\left(\theta, \alpha\right) = \arg\min_{\phi_i \in \mathbb{R}^d} \mathcal{L}\left(\theta, \alpha, \phi_i, \mathcal{D}_i^{\text{tr}}\right)$ of the inner level problem as the one SGD update from the meta-parameter $\hat{\phi}_i\left(\theta, \alpha\right) = \theta - \alpha \circ \frac{d}{d\phi}\mathcal{L}\left(\phi, \mathcal{D}_i^{\text{tr}}\right)|_{\phi=\theta}$ where $\circ$ denotes element-wise product. Then, Meta-SGD computes the gradient of the meta-loss $\frac{dF}{d\theta}$ of the outer level problem by approximating the meta-loss function $F$ as $\frac{1}{B} \sum_{i=1}^{B} \mathcal{L}\left(\hat{\phi}_i\left(\theta, \alpha\right); \mathcal{D}_i^{\text{test}}\right)$.

**Implicit MAML (iMAML)** (Rajeswaran et al., 2019) enforces the proximity of the task-specific parameters to the meta-parameters through a penalty term $\frac{\lambda}{2}\|\phi - \theta\|^2$. Thus, the target problem of iMAML can be modeled as following SLMF game:

$$\text{Leader:} \quad \min_{\theta \in \mathbb{R}^d} F(\theta, \phi) := \frac{1}{N} \sum_{i=1}^{N} \mathcal{L}\left(\theta, \phi_i; \mathcal{D}_i^{\text{test}}\right) \tag{17}$$

$$\text{Follower:} \quad \min_{\phi_i \in \mathbb{R}^d} \mathcal{L}\left(\theta, \phi_i; \mathcal{D}_i^{\text{tr}}\right) + \frac{\lambda}{2}\|\phi - \theta\|^2 \tag{18}$$

To compute a Stackelberg equilibrium of the SLMF game, iMAML first computes the optimal task-specific parameter $\phi_i^*(\theta) = \arg\min_{\phi_i \in \mathbb{R}^d} \mathcal{L}\left(\theta, \phi_i, \mathcal{D}_i^{\text{tr}}\right) + \frac{\lambda}{2}\|\phi - \theta\|^2$ of the inner level problem. Then, iMAML computes the gradient of the meta-loss $\frac{dF}{d\theta}$ of the outer level problem by approximating the meta-loss function $F$ as $\frac{1}{B} \sum_{i=1}^{B} \mathcal{L}\left(\phi_i^*(\theta); \mathcal{D}_i^{\text{test}}\right)$.

**CAVIA** (Zintgraf et al., 2019) learns input context parameters in the inner level and meta-parameter in the outer level. Thus, the target problem of CAVIA can be modeled as following SLMF game:

$$\text{Leader:} \quad \min_{\theta \in \mathbb{R}^d} F(\theta, \phi) := \frac{1}{N} \sum_{i=1}^{N} \mathcal{L}\left(\theta, \phi_i; \mathcal{D}_i^{\text{test}}\right) \tag{19}$$

$$\text{Follower:} \quad \min_{\phi_i \in \mathbb{R}^d} \mathcal{L}\left(\theta, \phi_i; \mathcal{D}_i^{\text{tr}}\right) \tag{20}$$

To compute a Stackelberg equilibrium of the SLMF game, CAVIA with a one-step gradient update first approximates the optimal task-specific parameter $\phi_i^*(\theta) = \arg\min_{\phi_i \in \mathbb{R}^n} \mathcal{L}\left(\theta, \phi_i, \mathcal{D}_i^{\text{tr}}\right)$ of the inner level problem as the one SGD update from the initial input context parameter $\hat{\phi}_i(\theta) = \phi_0 - \alpha \frac{d}{d\phi} \mathcal{L}\left(\theta, \phi; \mathcal{D}_i^{\text{tr}}\right)|_{\phi=\phi_0}$ where $\phi_0 \in \mathbb{R}^n$ is the initial input context parameter, and $n$ is a dimension of the input context parameter. Then, CAVIA computes the gradient of the meta-loss $\frac{dF}{d\theta}$ of the outer level problem by approximating the meta-loss function $F$ as $\frac{1}{B} \sum_{i=1}^{B} \mathcal{L}\left(\theta, \hat{\phi}_i(\theta); \mathcal{D}_i^{\text{test}}\right)$.

**TR-MAML** (Collins et al., 2020) learns task-robust meta-parameter. That is, the meta-parameter minimizes the worst-case loss. Thus, the target problem of TR-MAML can be modeled as following SLMF game:

$$\text{Leader:} \quad \min_{\theta \in \mathbb{R}^d} F(\theta, \phi) := \min_{\theta \in \mathbb{R}^d} \max_{i \in [N]} \mathcal{L}\left(\theta, \phi_i; \mathcal{D}_i^{\text{test}}\right) \tag{21}$$

$$\text{Follower:} \quad \min_{\phi_i \in \mathbb{R}^d} \mathcal{L}\left(\theta, \phi_i; \mathcal{D}_i^{\text{tr}}\right) \tag{22}$$

To compute a Stackelberg equilibrium of the SLMF game, TR-MAML with a one-step gradient update first approximates the optimal task-specific parameter $\phi_i^*(\theta) = \arg\min_{\phi_i \in \mathbb{R}^d} \mathcal{L}\left(\theta, \phi_i, \mathcal{D}_i^{\text{tr}}\right)$ of the inner level problem as the one SGD update from the meta-parameter $\hat{\phi}_i(\theta) = \theta - \alpha \frac{d}{d\phi} \mathcal{L}\left(\phi_i; \mathcal{D}_i^{\text{tr}}\right)|_{\phi=\theta}$. Then, TR-MAML computes the gradient of the meta-loss $\frac{dF}{d\theta}$ of the outer level problem by approximating the meta-loss function $F$ as $\min_{\theta \in \mathbb{R}^d} \max_{i \in [B]} \mathcal{L}\left(\phi_i(\theta); \mathcal{D}_i^{\text{test}}\right)$.

# B PRELIMINARIES OF GAME THEORY

The optimal meta-parameter $\theta^*$ is a general Stackelberg equilibrium of the SLMF game described in equations (10)-(11). At each meta-gradient update, the target problem of Nash-GBML is modeled as the following SLMF game $\Gamma = \left\langle \{1\}, [B], F, (f_i)_{i \in [B]}, \mathbb{R}^d, (\Omega_i)_{i \in [B]} \right\rangle$:

$$\theta^* = \arg\min_{\theta \in \mathbb{R}^d} F(\theta, \phi(\theta)) \tag{23}$$

$$\phi_i(\theta) = \arg\min_{\phi_i \in \Omega_i} f_i(\theta, \phi_i, \phi_{-i}(\theta)) \tag{24}$$

where $F$ is the outer level objective function, $f_i$ is the task loss function, and $B$ is a batch size. First, we compute the joint task-specific parameter $\phi$ through the well-known algorithm such as the proximal-decomposition algorithm (Scutari et al., 2012; Atzeni et al., 2013), regularized NI-function type method (Facchinei & Kanzow, 2010; Jo & Park, 2020), or the projected reflected gradient descent (PRGD) method (Malitsky, 2015). Next, we compute the meta-gradient $\frac{d}{d\theta}F$ to update the meta-parameter.

PRGD method effectively computes a solution of variational inequality for the inner level of Nash-GBML. Because the inner level of Nash-GBML is the Nash equilibrium problem (NEP) (Nash, 1950; 1951), we have to compute a Nash equilibrium of the meta-training tasks. Here, we discuss the relation between a Nash equilibrium and a solution of variational inequality. First, we make the following assumption in order to discuss the existence and uniqueness of Nash equilibrium.

**Assumption B.1** *Suppose the following holds:*

- *Task $i$'s strategy set $\Omega_i$ is closed and convex.*

- *Task $i$'s inner level loss function $f_i(\theta, \phi_i, \phi_{-i})$ is L-smooth for all task $i$, i.e.*

$$\left\| \frac{\partial}{\partial \phi_i} f_i \left( \phi_i^1, \phi_{-i}, \theta \right) - \frac{\partial}{\partial \phi_i} f_i \left( \phi_i^2, \phi_{-i}, \theta \right) \right\| \leq L \left\| \phi_i^1 - \phi_i^2 \right\|, \forall \phi_i^1, \phi_i^2 \qquad (25)$$

- *Task $i$'s inner level loss function $f_i(\theta, \phi_i, \phi_{-i})$ is strongly convex for all task $i$. It means that $f_i(\theta, \phi_i, \phi_{-i})$ is convex, and the partial gradient of loss function $\frac{\partial}{\partial \phi_i} f_i(\theta, \phi_i, \phi_{-i})$ is $\mu$-strongly monotone for all task $i$ with condition number $\kappa = L/\mu$. $\mu$-strongly monotonicity of the gradient of the task $i$'s loss function is represented as*

$$\left\langle \frac{\partial}{\partial \phi_i} f_i \left( \theta, \phi_i^1, \phi_{-i} \right) - \frac{\partial}{\partial \phi_i} f_i \left( \theta, \phi_i^2, \phi_{-i} \right), \phi_i^1 - \phi_i^2 \right\rangle \geq \mu \left\| \phi_i^1 - \phi_i^2 \right\|^2, \forall \phi_i^1, \phi_i^2 \quad (26)$$

Now, we discuss the existence and uniqueness of the solution for the variational inequality of equation (24), and it is also the unique Nash equilibrium. Then, we prove that the PRGD method always converges to the unique Nash equilibrium under Assumption B.1.

**Lemma B.2** *Let $G(\theta) = \left\langle [B], (f_i)_{i \in [B]}, (\Omega_i)_{i \in [B]} \right\rangle$ be an inner level problem of the SLMF game $\Gamma = \left\langle \{1\}, [B], F, (f_i)_{i \in [B]}, \mathbb{R}^d, (\Omega_i)_{i \in [B]} \right\rangle$ modeling the Nash-GBML algorithm when meta-parameter is $\theta$. Then, there is the unique variational equilibrium of $G(\theta)$, and it is also the unique Nash equilibrium of $G(\theta)$.*

*Proof.* By Assumption B.1, $G(\theta)$ has the unique variational equilibrium by Theorem 2.3.3 of (Facchinei & Pang, 2003), and it is also the unique Nash equilibrium by Proposition 1.4.2 of (Facchinei & Pang, 2003).

**Lemma B.3** *Let $G(\theta) = \left\langle [B], (f_i)_{i \in [B]}, (\Omega_i)_{i \in [B]} \right\rangle$ be an inner level problem of the SLMF game $\Gamma = \left\langle \{1\}, [B], F, (f_i)_{i \in [B]}, \mathbb{R}^d, (\Omega_i)_{i \in [B]} \right\rangle$ modeling the Nash-GBML algorithm when meta-parameter is $\theta$. Then, the PRGD method always converges to the unique Nash equilibrium of $G(\theta)$ under Assumption B.1.*

*Proof.* By Theorem 3.3 of (Malitsky, 2015), the PRGD method always converges to the unique variational equilibrium of $G(\theta)$. By Lemma B.2, the variational equilibrium computed by the PRGD method is also the unique Nash equilibrium of $G(\theta)$.

We compute the implicit gradient for the Stackelberg equilibrium of the SLMF game $\Gamma = \left\langle \{1\}, [B], F, (f_i)_{i \in [B]}, \mathbb{R}^d, (\Omega_i)_{i \in [B]} \right\rangle$ by transforming it into the $1 - 1 - 1$ Stackelberg game $\hat{\Gamma} = \left\langle f^1, f^2, f^3, \Omega^1, \Omega^2, \Omega^3 \right\rangle$ (Jo et al., 2023).

**Lemma B.4** *Let* $\Gamma = \left\langle \{1\}, [B], F, (f_i)_{i\in[B]}, \mathbb{R}^d, (\Omega_i)_{i\in[B]} \right\rangle$ *be the SLMF game modeling the Nash-GBML algorithm. The inner level loss function* $f_i$ *is defined as the sum of training loss function* $\mathcal{L}$ *and penalty term* $p$, *that is,* $f_i(\theta, \phi_i, \phi_{-i}) = \mathcal{L}(\phi_i; \mathcal{D}_i^{\mathrm{tr}}) + p(\theta, \phi_i, \phi_{-i})$. *Assume that* $\Omega_i = \mathbb{R}^d$. *Then, the implicit gradient* $\frac{d\phi^*(\theta)}{d\theta}$ *is as follows:*

$$\frac{d\phi^*(\theta)}{d\theta} = -\mathbf{P}(\theta, \phi^*(\theta))^{-1} \mathbf{Q}(\theta, \phi^*(\theta)) \tag{27}$$

*where*

$$\mathbf{P}(\theta, \phi^*(\theta)) = \left[ \frac{\partial}{\partial \phi} \left[ \frac{\partial \mathcal{L}(\phi_i^*(\theta); \mathcal{D}_i^{\mathrm{tr}})}{\partial \phi_i} \right]_{i\in[B]}^{\mathrm{T}} + \frac{\partial^2 p(\theta, \phi^*(\theta))}{\partial \phi^2} \right]$$

$$\mathbf{Q}(\theta, \phi^*(\theta)) = \frac{\partial}{\partial \theta} \left[ \frac{\partial p(\theta, \phi^*(\theta))}{\partial \phi} \right]^{\mathrm{T}} \tag{28}$$

*Proof.* Let $\hat{\Gamma} = \left\langle f^1, f^2, f^3, \Omega^1, \Omega^2, \Omega^3 \right\rangle$ be a $1-1-1$ Stackelberg game where $f^1 = F$ is a first leader's utility function and its decision variable $\theta$ in strategy set $\Omega^1 = \mathbb{R}^d$, $f^2 = \sum_{i\in[B]} \left( \frac{\partial f_i}{\partial \phi_i} \right) \left( \phi_i - \hat{\phi}_i \right)$ is a second leader's utility function and its decision variable $\phi$ in strategy set $\Omega^2 = \prod_{i\in[B]} \Omega_i$, and $f^3 = -\sum_{i\in[B]} \left( \frac{\partial f_i}{\partial \phi_i} \right) \left( \phi_i - \hat{\phi}_i \right)$ is a follower's utility function and its decision variable $\hat{\phi}$ in strategy set $\Omega^3 = \prod_{i\in[B]} \Omega_i$. Then, the Stackelberg equilibrium $\left( \theta^*, \phi^*(\theta), \hat{\phi}^*(\theta^*, \phi^*(\theta)) \right)$ of $1-1-1$ Stackelberg game $\hat{\Gamma}$ that satisfies equations (29) - (31) is also a Stackelberg equilibrium $(\theta^*, \phi^*(\theta))$ of the SLMF game $\Gamma$ (Jo et al., 2023).

$$\theta^* = \arg \min_{\theta \in \mathbb{R}^d} f^1(\theta, \phi^*(\theta)) \tag{29}$$

$$\phi^*(\theta) = \arg \min_{\phi \in \Omega^2(\theta)} f^2\left( \theta, \phi, \hat{\phi}^*(\theta, \phi) \right) \tag{30}$$

$$\hat{\phi}^*(\theta, \phi) = \arg \min_{\hat{\phi} \in \Omega^3(\theta)} f^3\left( \theta, \phi, \hat{\phi} \right) \tag{31}$$

Now we compute the implicit gradient when every follower of $\Gamma$ has an unconstrained strategy set, i.e., $\Omega_i = \mathbb{R}^d$. The derivative of $f^3$ with respect to $\hat{\phi}$ is a zero vector regardless of $\hat{\phi}$ if the following holds.

$$\frac{d}{d\hat{\phi}} f^3\left( \theta, \phi, \hat{\phi}^*(\theta, \phi) \right) = \left[ \frac{\partial \mathcal{L}(\phi_i; \mathcal{D}_i^{\mathrm{tr}})}{\partial \phi_i} + \frac{\partial p(\phi, \theta)}{\partial \phi_i} \right]_{i\in[B]}^{\mathrm{T}} = \mathbf{0}^{Bd\times 1} \tag{32}$$

The optimal $\phi^*(\theta)$ should satisfy equation (32). That is, the derivative of equation (32) with respect to $\theta$ is $\mathbf{0}^{Bd\times d}$.

$$\begin{aligned}
\frac{d}{d\theta} \frac{df^3}{d\hat{\phi}} &= \frac{d}{d\theta} \left[ \frac{\partial \mathcal{L}(\phi_i^*(\theta); \mathcal{D}_i^{\mathrm{tr}})}{\partial \phi_i} + \frac{\partial p(\theta, \phi^*(\theta))}{\partial \phi_i} \right]_{i\in[B]}^{\mathrm{T}} \\
&= \frac{\partial}{\partial \phi} \left[ \frac{\partial \mathcal{L}(\phi_i^*(\theta); \mathcal{D}_i^{\mathrm{tr}})}{\partial \phi_i} \right]_{i\in[B]}^{\mathrm{T}} \frac{d\phi^*(\theta)}{d\theta} \\
&\quad + \frac{\partial}{\partial \theta} \left[ \frac{\partial p(\theta, \phi^*(\theta))}{\partial \phi} \right]^{\mathrm{T}} + \frac{\partial^2 p(\theta, \phi^*(\theta))}{\partial \phi^2} \frac{d\phi^*(\theta)}{d\theta} \\
&= \mathbf{0}^{Bd\times d} \tag{33}
\end{aligned}$$

Therefore, the implicit gradient is as follows.

$$\frac{d\phi^*(\theta)}{d\theta} = -\left[ \frac{\partial}{\partial \phi} \left[ \frac{\partial \mathcal{L}(\phi_i^*(\theta); \mathcal{D}_i^{\mathrm{tr}})}{\partial \phi_i} \right]_{i\in[B]}^{\mathrm{T}} + \frac{\partial^2 p(\theta, \phi^*(\theta))}{\partial \phi^2} \right]^{-1} \frac{\partial}{\partial \theta} \left[ \frac{\partial p(\theta, \phi^*(\theta))}{\partial \phi} \right]^{\mathrm{T}} \tag{34}$$

The rest of our study focuses on proving the convergence of Nash-GBML where $\Omega_i = \mathbb{R}^d$ for all task $i$. Even if there are constraints on the strategy set, convergence is guaranteed using the implicit gradient computed by Lemma B.4 in a similar manner. To prove the convergence of Nash-GBML, we first define the approximated gradient of the outer level objective function $F$ using Lemma B.4.

$$\frac{\hat{d}}{d\theta}F\left(\theta, \hat{\phi}\right) = \frac{\partial}{\partial\theta}F\left(\theta, \hat{\phi}\right) + \frac{d\hat{\phi}}{d\theta} \times \frac{\partial}{\partial\phi}F\left(\theta, \hat{\phi}\right) \tag{35}$$

where $\hat{\phi}$ is joint estimated task-specific parameter and

$$\frac{d\hat{\phi}}{d\theta} = -\mathbf{P}\left(\theta, \hat{\phi}\right)^{-1} \mathbf{Q}\left(\theta, \hat{\phi}\right)$$

$$\mathbf{P}\left(\theta, \hat{\phi}\right) = \left[ \frac{\partial}{\partial\phi}\left[ \frac{\partial\mathcal{L}\left(\hat{\phi}_i; \mathcal{D}_i^{\mathrm{tr}}\right)}{\partial\phi_i} \right]^{\mathrm{T}}_{i\in[B]} + \frac{\partial^2 g\left(\theta, \hat{\phi}\right)}{\partial\phi^2} \right]$$

$$\mathbf{Q}\left(\theta, \hat{\phi}\right) = \frac{\partial}{\partial\theta}\left[ \frac{\partial g\left(\theta, \hat{\phi}\right)}{\partial\phi} \right]^{\mathrm{T}} \tag{36}$$

## C TIME AND SPACE COMPLEXITY

We define a $\delta$-accurate estimation of the optimal joint task-specific parameter $\phi^*$ and an $\epsilon$-accurate estimation of the approximated gradient of the outer level objective function $\frac{\hat{d}}{d\theta}F$.

**Definition C.1** *Let the joint task-specific parameter $\hat{\phi}$ be a solution estimated by the well-known computing algorithm (Facchinei & Kanzow, 2010; Scutari et al., 2012; Atzeni et al., 2013; Malitsky, 2015; Jo & Park, 2020). Then, $\hat{\phi}$ is a $\delta$-accurate estimation of the optimal joint task-specific parameter $\phi^*$ if it satisfies the following:*

$$\left\| \hat{\phi} - \phi^* \right\| \leq \delta \tag{37}$$

**Definition C.2** *Let $\frac{\hat{d}}{d\theta}F$ be an approximated gradient of the outer level objective function. Then, $\hat{h}_\theta$ is an $\epsilon$-accurate estimation of the outer level objective function if it satisfies the following:*

$$\left\| \frac{\hat{d}}{d\theta}F\left(\theta, \hat{\phi}\right) - \hat{h}_\theta\left(\theta, \hat{\phi}\right) \right\| \leq \epsilon \tag{38}$$

Note that the joint convergent task-specific parameter $\hat{\phi}$ of the Nash-GBML algorithm is a $\delta$-accurate estimation of the unique solution of variational inequality because the PRGD method computes the solution of the variational inequality (Malitsky, 2015). We show that $\hat{\phi}$ computed by the PRGD method is also a $\delta$-accurate estimation of the unique Nash equilibrium by Lemma B.3. Now we prove the convergence of the inner level problem of the SLMF game $\Gamma = \left\langle \{1\}, [B], F, (f_i)_{i\in[B]}, \mathbb{R}^d, (\Omega_i)_{i\in[B]} \right\rangle$ which model the Nash-GBML algorithm.

**Theorem C.3** *Let $D$ be the diameter of search space of the joint task-specific parameter $\phi = (\phi_i)_{i\in[B]}$ in the inner level problem (i.e. $\|\phi - \phi^*\| \leq D$). Suppose that the projected reflected gradient descent (PRGD) method (Malitsky, 2015) is used to compute $\delta$-accurate estimation of the optimal joint task-specific parameter $\hat{\phi} = \left(\hat{\phi}_i\right)_{i\in[B]}$ of the Nash equilibrium, which is the convergent point of the inner level of Nash-GBML algorithm. Under Assumption B.1, Nash-GBML algorithm computes $\hat{\phi}$ with $O\left(\kappa \log\left(D/\delta\right)\right)$ number of iterations, and only $O\left(Mem\left(\nabla\mathcal{L}_i\right)\kappa\log\left(D/\delta\right)\right)$ memory is required throughout where $\kappa$ is the condition number.*

*Proof.* By Theorem 3.3 of (Malitsky, 2015), the PRGD method converges to the $\delta$-accurate estimation of the optimal joint task-specific parameter in $n$ steps, described as follows:

$$\left\| \hat{\phi} - \phi^* (\theta) \right\|^2 \leq \gamma^n D^2 \tag{39}$$

where $\gamma = \frac{1 - 2a\mu + \sqrt{1 + 4a^2\mu^2}}{2}$. Under Assumption B.1, the number of gradient updates to compute $\delta$-accurate estimation of the optimal joint task-specific parameter $\hat{\phi}$ is $-\frac{2}{\log \gamma} \log (D/\delta)$. Since $a$ is proportional to $1/L$, and $-\frac{2}{\log \gamma}$ is proportional to $\frac{1}{a\mu}$, the number of gradient computations to compute $\hat{\phi}$ is bounded as $\kappa \log (D/\delta)$. Because we compute the implicit gradient of the Nash-GBML algorithm independently for each task-specific parameter, the memory usage is the same as that of the other GBML algorithms (Rajeswaran et al., 2019).

## D CONVERGENCE ANALYSIS

We introduce assumptions regarding the outer level objective function $F$ and task $i$'s loss function $f_i$.

**Assumption D.1** *We denote the optimal training loss function of task $i$ as $\mathcal{L}^* (\theta) = \mathcal{L} (\theta, \phi^* (\theta))$. Suppose the following holds:*

- *For any $\theta$, $\frac{\partial}{\partial \theta} F$ is Lipschitz continuous with respect to $\phi$ with constant $L_1 > 0$, i.e.*

$$\left\| \frac{\partial}{\partial \theta} F (\theta, \phi^1) - \frac{\partial}{\partial \theta} F (\theta, \phi^2) \right\| \leq L_1 \left\| \phi^1 - \phi^2 \right\|, \forall \phi^1, \phi^2 \tag{40}$$

- *For any $\phi$, $\frac{\partial}{\partial \theta} F$ is Lipschitz continuous with respect to $\theta$ with constant $\bar{L}_1 > 0$, i.e.*

$$\left\| \frac{\partial}{\partial \theta} F (\theta^1, \phi) - \frac{\partial}{\partial \theta} F (\theta^2, \phi) \right\| \leq \bar{L}_1 \left\| \theta^1 - \theta^2 \right\|, \forall \theta^1, \theta^2 \tag{41}$$

- *For any $\theta$, $\frac{\partial}{\partial \phi} F$ is Lipschitz continuous with respect to $\phi$ with constant $L_2 > 0$, i.e.*

$$\left\| \frac{\partial}{\partial \phi} F (\theta, \phi^1) - \frac{\partial}{\partial \phi} F (\theta, \phi^2) \right\| \leq L_2 \left\| \phi^1 - \phi^2 \right\|, \forall \phi^1, \phi^2 \tag{42}$$

- *For any $\phi$, $\frac{\partial}{\partial \phi} F$ is Lipschitz continuous with respect to $\theta$ with constant $\bar{L}_2 > 0$, i.e.*

$$\left\| \frac{\partial}{\partial \phi} F (\theta^1, \phi) - \frac{\partial}{\partial \phi} F (\theta^2, \phi) \right\| \leq \bar{L}_2 \left\| \theta^1 - \theta^2 \right\|, \forall \theta^1, \theta^2 \tag{43}$$

- *For any $\theta$, any $\phi$, we have $\left\| \frac{\partial F}{\partial \phi} \right\| \leq C_1$ for some constant $C_1 > 0$.*

- *For any $\theta$, any $\phi$, we have $\left\| \frac{\partial F}{\partial \theta} \right\| \leq \bar{C}_1$ for some constant $\bar{C}_1 > 0$.*

- *For any $\theta$, $\frac{\partial}{\partial \phi} \left[ \frac{\partial \mathcal{L}_{\mathcal{T}_i}}{\partial \phi_i} \right]_{i \in [B]}^{\mathrm{T}}$ is Lipschitz continuous with respect to $\phi$ with constant $L_3 > 0$ where $\phi = (\phi_i)_{i \in [B]}$, i.e.*

$$\left\| \frac{\partial}{\partial \phi} \left[ \frac{\partial \mathcal{L}_{\mathcal{T}_i} (\phi_i^1)}{\partial \phi_i} \right]_{i \in [B]}^{\mathrm{T}} - \frac{\partial}{\partial \phi} \left[ \frac{\partial \mathcal{L}_{\mathcal{T}_i} (\phi_i^2)}{\partial \phi_i} \right]_{i \in [B]}^{\mathrm{T}} \right\| \leq L_3 \left\| \phi^1 - \phi^2 \right\|, \forall \phi^1, \phi^2 \tag{44}$$

- *For any $\theta$, $\frac{\partial^2 p}{\partial \theta \partial \phi}$ is Lipschitz continuous with respect to $\phi$ with constant $L_4 > 0$, i.e.*

$$\left\| \frac{\partial^2}{\partial \theta \partial \phi} p (\theta, \phi^1) - \frac{\partial^2}{\partial \theta \partial \phi} p (\theta, \phi^2) \right\| \leq L_4 \left\| \phi^1 - \phi^2 \right\|, \forall \phi^1, \phi^2 \tag{45}$$

- *For any $\theta$, $\frac{\partial^2 p}{\partial \phi^2}$ is Lipschitz continuous with respect to $\phi$ with constant $L_5 > 0$, i.e.*

$$\left\| \frac{\partial^2}{\partial \phi^2} p\left(\theta, \phi^1\right) - \frac{\partial^2}{\partial \phi^2} p\left(\theta, \phi^2\right) \right\| \le L_5 \left\| \phi^1 - \phi^2 \right\|, \forall \phi^1, \phi^2 \tag{46}$$

- *The optimal loss function of task $i$, $\mathcal{L}_{\mathcal{T}_i}^*\left(\theta\right)$ is $L_6$-smooth for all task $i$, i.e.*

$$\mathcal{L}_{\mathcal{T}_i}^*\left(\theta^1\right) \le \mathcal{L}_{\mathcal{T}_i}^*\left(\theta^2\right) + \left\langle \theta^1 - \theta^2, \frac{d}{d\theta} \mathcal{L}_{\mathcal{T}_i}^*\left(\theta^2\right) \right\rangle + \frac{L_6}{2} \left\| \theta^1 - \theta^2 \right\|^2, \forall \theta^1, \theta^2 \tag{47}$$

- *For any $\theta$, $\mathcal{L}_{\mathcal{T}_i}$ is strongly convex with respect to $\phi_i$ with parameter $\mu_1 > 0$, i.e.*

$$\mu_1 \mathbf{I} \preceq \frac{\partial^2 \mathcal{L}_{\mathcal{T}_i}}{\partial \phi_i^2} \tag{48}$$

- *For any $\theta$, $p$ is strongly convex with respect to $\phi$ with parameter $\mu_2 > 0$, i.e.*

$$\mu_2 \mathbf{I} \preceq \frac{\partial^2 p}{\partial \phi^2} \tag{49}$$

- *For any $\theta$, $\mathcal{L}_{\mathcal{T}_i}\left(\theta, \phi^*\left(\theta\right)\right)$ is strongly convex on $\theta$.*

- *For any $\theta$, any $\phi$, we have $\left\| \frac{\partial^2 p}{\partial \theta \partial \phi} \right\| \le C_2$ for some constant $C_2 > 0$.*

- *Assumption B.1 is satisfied.*

The second main result is that the error between the estimated gradient $\hat{h}_\theta$ computed through back-propagation and $\frac{dF}{d\theta}$. The error is bounded by a weighted sum of the error in estimating $\phi$ and the error in estimating gradient through back-propagation.

**Lemma D.2** *Let $\theta$ be a given meta-parameter, $\phi^*$ be an optimal task-specific parameter, and $\hat{\phi}$ be an estimated task-specific parameter. Under Assumption D.1, the following statements hold.*

- *For the same sampling tasks, $\phi^*\left(\theta\right)$ is Lipschitz continuous with respect to $\theta$ with constant $\frac{C_2}{\mu_1+\mu_2} > 0$, i.e.*

$$\left\| \phi^*\left(\theta^1\right) - \phi^*\left(\theta^2\right) \right\| \le \frac{C_2}{\mu_1 + \mu_2} \left\| \theta^1 - \theta^2 \right\|, \forall \theta^1, \theta^2 \tag{50}$$

- *The difference between the approximated gradient $\frac{\hat{d}F}{d\theta}$ and $\frac{dF}{d\theta}$ is bounded by the error in estimating $\phi^*$. That is,*

$$\left\| \frac{d}{d\theta} F\left(\theta, \phi^*\left(\theta\right)\right) - \frac{\hat{d}}{d\theta} F\left(\theta, \hat{\phi}\left(\theta\right)\right) \right\| \le C \left\| \phi^*\left(\theta\right) - \hat{\phi}\left(\theta\right) \right\| \tag{51}$$

*where $C = L_1 + \frac{C_1 L_4 + C_2 L_2}{\mu_1 + \mu_2} + \frac{C_1 C_2 (L_3 + L_5)}{(\mu_1 + \mu_2)^2}$.*

- *The gradient of the optimal $F$ with respect to $\theta$ is Lipschitz continuous in $\theta$ with constant $L_F > 0$, i.e.*

$$\left\| \frac{d}{d\theta} F\left(\theta^1, \phi^*\left(\theta^1\right)\right) - \frac{d}{d\theta} F\left(\theta^2, \phi^*\left(\theta^2\right)\right) \right\| \le L_F \left\| \theta^1 - \theta^2 \right\| \tag{52}$$

*where $L_F = \bar{L}_1 + \frac{C_2}{\mu_1 + \mu_2}\left(\bar{L}_2 + C\right)$.*

*Proof.* First, we prove the implicit gradient is bounded. The implicit gradient $\frac{d\phi^*(\theta)}{d\theta}$ is computed as follows by Lemma B.4.

$$\frac{d\phi^*\left(\theta\right)}{d\theta} = -\mathbf{P}\left(\phi^*\left(\theta\right), \theta\right)^{-1} \mathbf{Q}\left(\phi^*\left(\theta\right), \theta\right) \tag{53}$$

Under Assumption D.1, $\mathbf{P}^{-1}$ and $\mathbf{Q}$ is bounded as follows.

$$\left\| \mathbf{P}\left(\phi^*\left(\theta\right),\theta\right)^{-1} \right\| \leq \frac{1}{\mu_1 + \mu_2}$$
$$\left\| \mathbf{Q}\left(\phi^*\left(\theta\right),\theta\right) \right\| \leq C_2 \tag{54}$$

Thus, the implicit gradient is bounded as $\frac{C_2}{\mu_1+\mu_2}$.

$$\begin{aligned}
\left\| \frac{d\phi^*\left(\theta\right)}{d\theta} \right\| &= \left\| -\mathbf{P}\left(\phi^*\left(\theta\right),\theta\right)^{-1}\mathbf{Q}\left(\phi^*\left(\theta\right),\theta\right) \right\| \\
&\leq \left\| \mathbf{P}\left(\phi^*\left(\theta\right),\theta\right)^{-1} \right\| \times \left\| \mathbf{Q}\left(\phi^*\left(\theta\right),\theta\right) \right\| \\
&\leq \frac{C_2}{\mu_1 + \mu_2}
\end{aligned} \tag{55}$$

Now we can prove $\phi^*\left(\theta\right)$ is Lipschitz continuous. The following holds for all $\theta^1, \theta^2$.

$$\begin{aligned}
\left\| \phi^*\left(\theta^1\right) - \phi^*\left(\theta^2\right) \right\| &\leq \left\| \frac{d\phi^*\left(\theta\right)}{d\theta} \right\| \left\| \theta^1 - \theta^2 \right\| \\
&\leq \frac{C_2}{\mu_1 + \mu_2}\left\| \theta^1 - \theta^2 \right\|
\end{aligned} \tag{56}$$

Next, we prove the difference between the approximated gradient $\frac{\hat{d}F}{d\theta}$ and $\frac{dF}{d\theta}$ is bounded.

$$\begin{aligned}
\left\| \frac{d}{d\theta}F\left(\theta,\phi^*\left(\theta\right)\right) - \frac{\hat{d}}{d\theta}F\left(\theta,\hat{\phi}\left(\theta\right)\right) \right\| &= \left\| \mathbf{M}_1 + \mathbf{M}_2 \right\| \\
&= \left\| \mathbf{M}_1 + \mathbf{M}_3 + \mathbf{M}_4 \right\| \\
&= \left\| \mathbf{M}_1 + \mathbf{M}_3 + \left(\mathbf{M}_5 + \mathbf{M}_6\right)\frac{\partial}{\partial\phi}F\left(\theta,\phi^*\left(\theta\right)\right) \right\|
\end{aligned} \tag{57}$$

where

$$\begin{aligned}
\mathbf{M}_1 =& \frac{\partial}{\partial\theta}F\left(\theta,\phi^*\left(\theta\right)\right) - \frac{\partial}{\partial\theta}F\left(\theta,\hat{\phi}\left(\theta\right)\right) \\
\mathbf{M}_2 =& \frac{d\phi^*\left(\theta\right)}{d\theta}\frac{\partial}{\partial\phi}F\left(\theta,\phi^*\left(\theta\right)\right) - \frac{d\hat{\phi}\left(\theta\right)}{d\theta}\frac{\partial}{\partial\phi}F\left(\theta,\hat{\phi}\left(\theta\right)\right) \\
\mathbf{M}_3 =& \frac{d\hat{\phi}\left(\theta\right)}{d\theta}\left(\frac{\partial}{\partial\phi}F\left(\theta,\phi^*\left(\theta\right)\right) - \frac{\partial}{\partial\phi}F\left(\theta,\hat{\phi}\left(\theta\right)\right)\right) \\
\mathbf{M}_4 =& \left(\frac{d\phi^*\left(\theta\right)}{d\theta} - \frac{d\hat{\phi}\left(\theta\right)}{d\theta}\right)\frac{\partial}{\partial\phi}F\left(\theta,\phi^*\left(\theta\right)\right) \\
\mathbf{M}_5 =& \left(\mathbf{P}\left(\hat{\phi}\left(\theta\right),\theta\right)^{-1} - \mathbf{P}\left(\phi^*\left(\theta\right),\theta\right)^{-1}\right)\mathbf{Q}\left(\phi^*\left(\theta\right),\theta\right) \\
\mathbf{M}_6 =& \mathbf{P}\left(\hat{\phi}\left(\theta\right),\theta\right)^{-1}\left(\mathbf{Q}\left(\hat{\phi}\left(\theta\right),\theta\right) - \mathbf{Q}\left(\phi^*\left(\theta\right),\theta\right)\right)
\end{aligned} \tag{58}$$

Under Assumption D.1, each term of equation (57) satisfies the following inequalities. Because $\frac{\partial}{\partial\theta}F$ is Lipschitz continuous,

$$\left\| \mathbf{M}_1 \right\| \leq L_1\left\| \phi^*\left(\theta\right) - \hat{\phi}\left(\theta\right) \right\| \tag{59}$$

Because $\frac{\partial}{\partial \phi} F$ is Lipschitz continuous,

$$
\begin{aligned}
\|\mathbf{M}_3\| &\leq \left\| \frac{d\hat{\phi}(\theta)}{d\theta} \right\| \left\| \frac{\partial}{\partial \phi} F(\theta, \phi^*(\theta)) - \frac{\partial}{\partial \phi} F\left(\theta, \hat{\phi}(\theta)\right) \right\| \\
&\leq \left\| -\mathbf{P}\left(\hat{\phi}(\theta), \theta\right)^{-1} \mathbf{Q}\left(\hat{\phi}(\theta), \theta\right) \right\| \times L_2 \left\| \phi^*(\theta) - \hat{\phi}(\theta) \right\| \\
&\leq \frac{C_2 L_2}{\mu_1 + \mu_2} \left\| \phi^*(\theta) - \hat{\phi}(\theta) \right\|
\end{aligned}
\tag{60}
$$

Because $\mathbf{P}$ is Lipschitz continuous,

$$
\begin{aligned}
\|\mathbf{M}_5\| &\leq \left\| \mathbf{P}\left(\hat{\phi}(\theta), \theta\right)^{-1} - \mathbf{P}(\phi^*(\theta), \theta)^{-1} \right\| \left\| \frac{\partial^2 p}{\partial\theta\partial\phi} \right\| \\
&= \left\| \mathbf{P}(\phi^*(\theta), \theta)^{-1} \left( \mathbf{P}(\phi^*(\theta), \theta) - \mathbf{P}\left(\hat{\phi}(\theta), \theta\right) \right) \mathbf{P}\left(\hat{\phi}(\theta), \theta\right)^{-1} \right\| \left\| \frac{\partial^2 p}{\partial\theta\partial\phi} \right\| \\
&\leq \left\| \mathbf{P}(\phi^*(\theta), \theta)^{-1} \right\| \left\| \mathbf{P}(\phi^*(\theta), \theta) - \mathbf{P}\left(\hat{\phi}(\theta), \theta\right) \right\| \left\| \mathbf{P}\left(\hat{\phi}(\theta), \theta\right)^{-1} \right\| \left\| \frac{\partial^2 p}{\partial\theta\partial\phi} \right\| \\
&\leq \frac{C_2}{(\mu_1 + \mu_2)^2} \left\| \frac{\partial}{\partial\phi} \left[ \frac{\partial \mathcal{L}_{\mathcal{T}_i}(\phi_i^*(\theta))}{\partial\phi_i} \right]^{\mathrm{T}}_{i\in[B]} - \frac{\partial}{\partial\phi} \left[ \frac{\partial \mathcal{L}_{\mathcal{T}_i}\left(\hat{\phi}_i(\theta)\right)}{\partial\phi_i} \right]^{\mathrm{T}}_{i\in[B]} \right\| \\
&\quad + \frac{C_2}{(\mu_1 + \mu_2)^2} \left\| \frac{\partial^2 p(\phi^*(\theta), \theta)}{\partial\phi^2} - \frac{\partial^2 p\left(\hat{\phi}(\theta), \theta\right)}{\partial\phi^2} \right\| \\
&\leq \frac{C_2(L_3 + L_5)}{(\mu_1 + \mu_2)^2} \left\| \phi^*(\theta) - \hat{\phi}(\theta) \right\|
\end{aligned}
\tag{61}
$$

Because $\frac{\partial^2 p}{\partial\theta\partial\phi}$ is Lipschitz continuous,

$$
\begin{aligned}
\left\| (\mathbf{M}_5 + \mathbf{M}_6) \frac{\partial}{\partial\phi} F(\theta, \phi^*(\theta)) \right\| &\leq (\|\mathbf{M}_5\| + \|\mathbf{M}_6\|) \left\| \frac{\partial}{\partial\phi} F(\theta, \phi^*(\theta)) \right\| \\
&\leq \frac{C_1 C_2 (L_3 + L_5)}{(\mu_1 + \mu_2)^2} \left\| \phi^*(\theta) - \hat{\phi}(\theta) \right\| \\
&\quad + \frac{L_4 C_1}{\mu_1 + \mu_2} \left\| \phi^*(\theta) - \hat{\phi}(\theta) \right\|
\end{aligned}
\tag{62}
$$

Thus, the equation (57) is expanded as follows.

$$
\begin{aligned}
\left\| \frac{d}{d\theta} F(\theta, \phi^*(\theta)) - \frac{\hat{d}}{d\theta} F\left(\theta, \hat{\phi}(\theta)\right) \right\| &\leq \left\| \mathbf{M}_1 + \mathbf{M}_3 + (\mathbf{M}_5 + \mathbf{M}_6) \frac{\partial}{\partial\phi} F(\theta, \phi^*(\theta)) \right\| \\
&\leq \|\mathbf{M}_1\| + \|\mathbf{M}_3\| + \left\| (\mathbf{M}_5 + \mathbf{M}_6) \frac{\partial}{\partial\phi} F(\theta, \phi^*(\theta)) \right\| \\
&\leq C \left\| \phi^*(\theta) - \hat{\phi}(\theta) \right\|
\end{aligned}
\tag{63}
$$

where

$$
C = L_1 + \frac{C_1 L_4 + C_2 L_2}{\mu_1 + \mu_2} + \frac{C_1 C_2 (L_3 + L_5)}{(\mu_1 + \mu_2)^2}
\tag{64}
$$

Finally, we prove the gradient of the optimal $F$ with respect to $\theta$ is Lipschitz continuous in $\theta$.

$$
\left\| \frac{d}{d\theta} F \left( \theta^1, \phi^* \left( \theta^1 \right) \right) - \frac{d}{d\theta} F \left( \theta^2, \phi^* \left( \theta^2 \right) \right) \right\| \leq \left\| \frac{d}{d\theta} F \left( \theta^1, \phi^* \left( \theta^1 \right) \right) - \frac{\hat{d}}{d\theta} F \left( \theta^1, \phi^* \left( \theta^2 \right) \right) \right\|
$$

$$
+ \left\| \frac{\hat{d}}{d\theta} F \left( \theta^1, \phi^* \left( \theta^2 \right) \right) - \frac{d}{d\theta} F \left( \theta^2, \phi^* \left( \theta^2 \right) \right) \right\| \quad (65)
$$

The first term of equation (65) is expanded as follows using equations (50), (51).

$$
\left\| \frac{d}{d\theta} F \left( \theta^1, \phi^* \left( \theta^1 \right) \right) - \frac{\hat{d}}{d\theta} F \left( \theta^1, \phi^* \left( \theta^2 \right) \right) \right\| \leq C \left\| \phi^* \left( \theta^1 \right) - \phi^* \left( \theta^2 \right) \right\|
$$

$$
\leq \frac{C C_2}{\mu_1 + \mu_2} \left\| \theta^1 - \theta^2 \right\| \quad (66)
$$

Under Assumption D.1, the second term of equation (65) is expanded as

$$
\left\| \frac{\hat{d}}{d\theta} F \left( \theta^1, \phi^* \left( \theta^2 \right) \right) - \frac{d}{d\theta} F \left( \theta^2, \phi^* \left( \theta^2 \right) \right) \right\| \leq \left\| \frac{\partial}{\partial\theta} F \left( \theta^1, \phi^* \left( \theta^2 \right) \right) - \frac{\partial}{\partial\theta} F \left( \theta^2, \phi^* \left( \theta^2 \right) \right) \right\|
$$

$$
+ \left\| \frac{d\phi^* \left( \theta^2 \right)}{d\theta} \right\| \left\| \frac{\partial}{\partial\phi} F \left( \theta^1, \phi^* \left( \theta^2 \right) \right) - \frac{\partial}{\partial\phi} F \left( \theta^2, \phi^* \left( \theta^2 \right) \right) \right\|
$$

$$
\leq \bar{L}_1 \left\| \theta^1 - \theta^2 \right\| + \frac{C_2}{\mu_1 + \mu_2} \bar{L}_2 \left\| \theta^1 - \theta^2 \right\| \quad (67)
$$

Now we prove the optimal $F^*$ is Lipschitz continuous with respect to $\theta$ with constant $L_F$ by equations (66) and (67).

$$
\left\| \frac{d}{d\theta} F \left( \theta^1, \phi^* \left( \theta^1 \right) \right) - \frac{d}{d\theta} F \left( \theta^2, \phi^* \left( \theta^2 \right) \right) \right\| \leq L_F \left\| \theta^1 - \theta^2 \right\| \quad (68)
$$

where $L_F = \bar{L}_1 + \frac{C_2}{\mu_1 + \mu_2} \left( \bar{L}_2 + C \right)$.

**Theorem D.3** *Let $\theta$ be a given meta-parameter, $\phi^*$ be an optimal task-specific parameter, $\hat{\phi}$ be a $\delta$-accurate estimated task-specific parameter, and $\hat{h}_\theta$ be an $\epsilon$-accurate estimated gradient of $F$ with respect to $\theta$ computed through back-propagation. Under Assumption D.1, the difference between the $\epsilon$-accurate estimated gradient $\hat{h}_\theta$ and the gradient of the optimal outer level objective function $F$ with respect to $\theta$, $\frac{dF}{d\theta}$, is bounded by the weighted sum of the error in estimating $\phi^*$ and the error in estimating the gradient through back-propagation. That is,*

$$
\left\| \frac{d}{d\theta} F \left( \theta, \phi^* \left( \theta \right) \right) - \hat{h}_\theta \left( \theta, \hat{\phi} \right) \right\| \leq C \left\| \phi^* \left( \theta \right) - \hat{\phi} \right\| + \left\| \frac{\hat{d}}{d\theta} F \left( \theta, \hat{\phi} \right) - \hat{h}_\theta \left( \theta, \hat{\phi} \right) \right\|
$$

$$
\leq C\delta + \epsilon \quad (69)
$$

*where $C = L_1 + \frac{C_1 L_4 + C_2 L_2}{\mu_1 + \mu_2} + \frac{C_1 C_2 (L_3 + L_5)}{(\mu_1 + \mu_2)^2}$.*

*Proof.* Because the triangle inequality $\| a + b \| \leq \| a \| + \| b \|$ holds,

$$
\left\| \frac{d}{d\theta} F \left( \theta, \phi^* \left( \theta \right) \right) - \hat{h}_\theta \left( \theta, \hat{\phi} \right) \right\| = \left\| \frac{d}{d\theta} F \left( \theta, \phi^* \left( \theta \right) \right) - \frac{\hat{d}}{d\theta} F \left( \theta, \hat{\phi} \right) + \frac{\hat{d}}{d\theta} F \left( \theta, \hat{\phi} \right) - \hat{h}_\theta \left( \theta, \hat{\phi} \right) \right\|
$$

$$
\leq \left\| \frac{d}{d\theta} F \left( \theta, \phi^* \left( \theta \right) \right) - \frac{\hat{d}}{d\theta} F \left( \theta, \hat{\phi} \right) \right\| + \left\| \frac{\hat{d}}{d\theta} F \left( \theta, \hat{\phi} \right) - \hat{h}_\theta \left( \theta, \hat{\phi} \right) \right\|
$$

$$
\leq \left\| \frac{d}{d\theta} F \left( \theta, \phi^* \left( \theta \right) \right) - \frac{\hat{d}}{d\theta} F \left( \theta, \hat{\phi} \right) \right\| + \epsilon \quad (70)
$$

By Lemma D.2, equation (70) is expanded as follows.

$$\left\| \frac{d}{d\theta} F\left(\theta, \phi^*\left(\theta\right)\right) - \hat{h}_\theta\left(\theta, \hat{\phi}\right) \right\| \leq \left\| \frac{d}{d\theta} F\left(\theta, \phi^*\left(\theta\right)\right) - \frac{\hat{d}}{d\theta} F\left(\theta, \hat{\phi}\right) \right\| + \epsilon$$

$$\leq C\left\| \phi^*\left(\theta\right) - \hat{\phi} \right\| + \epsilon$$

$$\leq C\delta + \epsilon \tag{71}$$

where $C = L_1 + \frac{C_1 L_4 + C_2 L_2}{\mu_1 + \mu_2} + \frac{C_1 C_2 (L_3 + L_5)}{(\mu_1 + \mu_2)^2}$.

For the rest of our paper, we discuss the expected utility function with respect to task sampling in order to prove the convergence of the Nash-GBML algorithm, and its convergent point is an optimal solution of the stochastic optimization problem described in equations (10) and (11). First, we describe the assumptions and lemmas required to prove the convergence. We denote $\mathbb{E}_{\mathcal{T}_i \sim p(\mathcal{T})}\left[\cdot\right]$ as $\mathbb{E}\left[\cdot\right]$ in the remaining part.

**Assumption D.4 (Convexity assumption)** *Let $\theta^{(t)}$ be a meta-parameter, $\phi^*\left(\theta^{(t)}\right)$ be an optimal joint task-specific parameter, and $\hat{\phi}\left(\theta^{(t)}\right)$ be an estimated joint task-specific parameter of the $t$-th updated meta-parameter. For any $k \geq 0$, there exists a non-increasing sequence $\{b_k\}_{k\geq 0}$, $\{\sigma_k\}_{k\geq 0}$, and $\{\bar{\sigma}_k\}_{k\geq 0}$ which converge to $0$ that satisfies the following.*

- *Let $\hat{\phi}$ be an estimated joint task-specific parameter. Then, the expectation of an estimated gradient $\hat{h}_\theta\left(\theta, \hat{\phi}\left(\theta\right)\right)$ is as follows.*

$$\mathbb{E}\left[\hat{h}_\theta\left(\theta^{(t)}, \hat{\phi}\left(\theta^{(t)}\right)\right)\right] = \mathbb{E}\left[\frac{\hat{d}}{d\theta} F\left(\theta^{(t)}, \hat{\phi}\left(\theta^{(t)}\right)\right)\right] + B_t, \|B_t\| \leq b_t \tag{72}$$

- *The norm-variance of an estimated gradient $\hat{h}_\theta\left(\theta, \hat{\phi}\left(\theta\right)\right)$ is bounded, i.e.*

$$\mathbb{E}\left[\left\| \hat{h}_\theta\left(\theta^{(t)}, \hat{\phi}\left(\theta^{(t)}\right)\right) - \mathbb{E}\left[\hat{h}_\theta\left(\theta^{(t)}, \hat{\phi}\left(\theta^{(t)}\right)\right)\right] \right\|^2\right] \leq \sigma_t^2 \tag{73}$$

- *The norm-variance of an optimal gradient $\frac{d}{d\theta} F\left(\theta^{(t)}, \phi^*\left(\theta^{(t)}\right)\right)$ is bounded, i.e.*

$$\mathbb{E}\left[\left\| \frac{d}{d\theta} F\left(\theta^{(t)}, \phi^*\left(\theta^{(t)}\right)\right) - \mathbb{E}\left[\frac{d}{d\theta} F\left(\theta^{(t)}, \phi^*\left(\theta^{(t)}\right)\right)\right] \right\|^2\right] \leq \bar{\sigma}_t^2 \tag{74}$$

- *Assumption D.1 is satisfied.*

**Lemma D.5** *Under Assumption D.4, the expectation of the square norm of an estimated gradient $\hat{h}_\theta\left(\theta, \hat{\phi}\left(\theta\right)\right)$ is bounded, i.e.*

$$\mathbb{E}\left[\left\| \hat{h}_\theta\left(\theta^{(t)}, \hat{\phi}\left(\theta^{(t)}\right)\right) \right\|^2\right] \leq 4C^2 \mathbb{E}\left[\left\| \hat{\phi}\left(\theta^{(t)}\right) - \phi^*\left(\theta^{(t)}\right) \right\|\right]^2$$

$$+ \sigma_t^2 + 2b_t^2 + 4\left(\bar{C}_1 + \frac{C_1 C_2}{\mu_1 + \mu_2}\right)^2 \tag{75}$$

*where $C = L_1 + \frac{C_1 L_4 + C_2 L_2}{\mu_1 + \mu_2} + \frac{C_1 C_2 (L_3 + L_5)}{(\mu_1 + \mu_2)^2}$.*

*Proof.* We denote $\hat{h}_\theta\left(\theta, \hat{\phi}(\theta)\right)$ as $\hat{h}_\theta(\theta)$. Then, the expectation of the square norm of the estimated gradient $\hat{h}_\theta(\theta)$ is as follows.

$$
\begin{aligned}
\mathbb{E}\left[\left\|\hat{h}_\theta(\theta)\right\|^2\right] &= \mathbb{E}\left[\left\|\hat{h}_\theta(\theta)\right\|^2\right] + 2\left\|\mathbb{E}\left[\hat{h}_\theta(\theta)\right]\right\|^2 - 2\left\langle\mathbb{E}\left[\hat{h}_\theta(\theta)\right], \mathbb{E}\left[\hat{h}_\theta(\theta)\right]\right\rangle \\
&= \mathbb{E}\left[\left\|\hat{h}_\theta(\theta)\right\|^2\right] + 2\left\|\mathbb{E}\left[\hat{h}_\theta(\theta)\right]\right\|^2 - 2\mathbb{E}\left\langle\hat{h}_\theta(\theta), \mathbb{E}\left[\hat{h}_\theta(\theta)\right]\right\rangle \\
&= \mathbb{E}\left[\left\|\hat{h}_\theta(\theta) - \mathbb{E}\left[\hat{h}_\theta(\theta)\right]\right\|^2\right] + \left\|\mathbb{E}\left[\hat{h}_\theta(\theta)\right]\right\|^2
\end{aligned}
\tag{76}
$$

Substituting $\theta$ with $\theta^{(t)}$ of equation (76). Because $\|a+b\|^2 \leq 2\|a\|^2 + 2\|b\|^2$, equation (76) is expanded as follows Under Assumption D.4.

$$
\begin{aligned}
\mathbb{E}\left[\left\|\hat{h}_\theta\left(\theta^{(t)}\right)\right\|^2\right] &= \mathbb{E}\left[\left\|\hat{h}_\theta\left(\theta^{(t)}\right) - \mathbb{E}\left[\hat{h}_\theta\left(\theta^{(t)}\right)\right]\right\|^2\right] + \left\|\mathbb{E}\left[\hat{h}_\theta\left(\theta^{(t)}\right)\right]\right\|^2 \\
&\leq \sigma_t^2 + \left\|\mathbb{E}\left[\frac{\hat{d}}{d\theta}F\left(\theta^{(t)}, \hat{\phi}\left(\theta^{(t)}\right)\right)\right] + B_t\right\|^2 \\
&\leq \sigma_t^2 + 2b_t^2 + 2\left\|\mathbb{E}\left[\frac{\hat{d}}{d\theta}F\left(\theta^{(t)}, \hat{\phi}\left(\theta^{(t)}\right)\right)\right]\right\|^2 \\
&\leq \sigma_t^2 + 2b_t^2 + 4\left\|\mathbb{E}\left[\frac{d}{d\theta}F\left(\theta^{(t)}, \phi^*\left(\theta^{(t)}\right)\right)\right]\right\|^2 \\
&\quad + 4\left\|\mathbb{E}\left[\frac{\hat{d}}{d\theta}F\left(\theta^{(t)}, \hat{\phi}\left(\theta^{(t)}\right)\right)\right] - \mathbb{E}\left[\frac{d}{d\theta}F\left(\theta^{(t)}, \phi^*\left(\theta^{(t)}\right)\right)\right]\right\|^2
\end{aligned}
\tag{77}
$$

Because the norm is convex, $\|\mathbb{E}[\cdot]\|^2 \leq \mathbb{E}[\|\cdot\|]^2$ by Jensen's inequality.

$$
\begin{aligned}
\mathbb{E}\left[\left\|\hat{h}_\theta\left(\theta^{(t)}\right)\right\|^2\right] &\leq \sigma_t^2 + 2b_t^2 + 4\left\|\mathbb{E}\left[\frac{d}{d\theta}F\left(\theta^{(t)}, \phi^*\left(\theta^{(t)}\right)\right)\right]\right\|^2 \\
&\quad + 4\left\|\mathbb{E}\left[\frac{\hat{d}}{d\theta}F\left(\theta^{(t)}, \hat{\phi}\left(\theta^{(t)}\right)\right) - \frac{d}{d\theta}F\left(\theta^{(t)}, \phi^*\left(\theta^{(t)}\right)\right)\right]\right\|^2 \\
&\leq \sigma_t^2 + 2b_t^2 + 4\mathbb{E}\left[\left\|\frac{d}{d\theta}F\left(\theta^{(t)}, \phi^*\left(\theta^{(t)}\right)\right)\right\|\right]^2 \\
&\quad + 4\mathbb{E}\left[\left\|\frac{\hat{d}}{d\theta}F\left(\theta^{(t)}, \hat{\phi}\left(\theta^{(t)}\right)\right) - \frac{d}{d\theta}F\left(\theta^{(t)}, \phi^*\left(\theta^{(t)}\right)\right)\right\|\right]^2
\end{aligned}
\tag{78}
$$

Because $\frac{dF}{d\theta}(\theta, \phi^*(\theta)) = \frac{\partial}{\partial\theta}F(\theta, \phi^*(\theta)) + \frac{d\phi^*(\theta)}{d\theta} \times \frac{\partial}{\partial\phi}F(\theta, \phi^*(\theta))$, the third term and fourth term of equation (78) are expanded as follows by Lemma D.2 and Assumption D.1.

$$
\begin{aligned}
\mathbb{E}\left[\left\|\frac{d}{d\theta}F\left(\theta^{(t)}, \phi^*\left(\theta^{(t)}\right)\right)\right\|\right] &\leq \mathbb{E}\left[\left\|\frac{\partial}{\partial\theta}F\left(\theta^{(t)}, \phi^*\left(\theta^{(t)}\right)\right)\right\|\right] \\
&\quad + \mathbb{E}\left[\left\|\frac{d\phi^*\left(\theta^{(t)}\right)}{d\theta}\right\|\left\|\frac{\partial}{\partial\phi}F\left(\theta^{(t)}, \phi^*\left(\theta^{(t)}\right)\right)\right\|\right] \\
&\leq \bar{C}_1 + \frac{C_1 C_2}{\mu_1 + \mu_2}
\end{aligned}
\tag{79}
$$

$$
\mathbb{E}\left[\left\|\frac{\hat{d}}{d\theta}F\left(\theta^{(t)}, \hat{\phi}\left(\theta^{(t)}\right)\right) - \frac{d}{d\theta}F\left(\theta^{(t)}, \phi^*\left(\theta^{(t)}\right)\right)\right\|\right] \leq C\mathbb{E}\left[\left\|\hat{\phi}\left(\theta^{(t)}\right) - \phi^*\left(\theta^{(t)}\right)\right\|\right]
\tag{80}
$$

where $C = L_1 + \frac{C_1 L_4 + C_2 L_2}{\mu_1 + \mu_2} + \frac{C_1 C_2 (L_3 + L_5)}{(\mu_1 + \mu_2)^2}$. Combining equations (78), (79) and (80), we derive the bound of the expectation of the estimated gradient.

$$
\mathbb{E}\left[\left\|\hat{h}_\theta\left(\theta^{(t)}\right)\right\|^2\right] \leq \sigma_t^2 + 2b_t^2 + 4\mathbb{E}\left[\left\|\frac{d}{d\theta}F\left(\theta^{(t)}, \phi^*\left(\theta^{(t)}\right)\right)\right\|\right]^2
$$

$$
+ 4\mathbb{E}\left[\left\|\frac{\hat{d}}{d\theta}F\left(\theta^{(t)}, \hat{\phi}\left(\theta^{(t)}\right)\right) - \frac{d}{d\theta}F\left(\theta^{(t)}, \phi^*\left(\theta^{(t)}\right)\right)\right\|\right]^2
$$

$$
\leq \sigma_t^2 + 2b_t^2 + 4\left(\bar{C}_1 + \frac{C_1 C_2}{\mu_1 + \mu_2}\right)^2 + 4C^2 \mathbb{E}\left[\left\|\hat{\phi}\left(\theta^{(t)}\right) - \phi^*\left(\theta^{(t)}\right)\right\|\right]^2 \tag{81}
$$

In this study, we prove the case where the outer level objective function is the average loss (Finn et al., 2017; Li et al., 2017; Rajeswaran et al., 2019; Zintgraf et al., 2019). We can similarly prove the case where the outer level objective function is the worst-case loss (Collins et al., 2020).

**Lemma D.6** Let $\mathcal{L}(\theta)$ be an expected optimal outer level objective function, that is, $\mathcal{L}(\theta) = \mathbb{E}[F(\theta, \phi^*(\theta))]$. Under Assumption D.4, $\mathcal{L}$ satisfies the following equation.

$$
\mathcal{L}\left(\theta^{(t+1)}\right) - \mathcal{L}\left(\theta^{(t)}\right) \leq \left(\frac{L_6}{2} - \frac{1}{2\beta}\right)\mathbb{E}\left[\left\|\theta^{(t+1)} - \theta^{(t)}\right\|^2\right]
$$

$$
+ 4\beta C^2 \mathbb{E}\left[\left\|\phi^*\left(\theta^{(t)}\right) - \hat{\phi}\left(\theta^{(t)}\right)\right\|\right]^2 + 4\beta\bar{\sigma}_t^2 + 2\beta b_t^2 + \beta\sigma_t^2 \tag{82}
$$

where $C = L_1 + \frac{C_1 L_4 + C_2 L_2}{\mu_1 + \mu_2} + \frac{C_1 C_2 (L_3 + L_5)}{(\mu_1 + \mu_2)^2}$.

*Proof.* Under Assumption D.1, the task $i$'s optimal loss function $\mathcal{L}^*_{\mathcal{T}_i}(\theta) = \mathcal{L}_{\mathcal{T}_i}(\theta, \phi^*(\theta))$ is $L_6$-smooth.

$$
\mathcal{L}^*_{\mathcal{T}_i}\left(\theta^{(t+1)}\right) \leq \mathcal{L}^*_{\mathcal{T}_i}\left(\theta^{(t)}\right) + \left\langle\theta^{(t+1)} - \theta^{(t)}, \frac{d}{d\theta}\mathcal{L}^*_{\mathcal{T}_i}\left(\theta^{(t)}\right)\right\rangle + \frac{L_6}{2}\left\|\theta^{(t+1)} - \theta^{(t)}\right\|^2 \tag{83}
$$

Summing up equation (83) for all tasks $i \in [B]$ and divide it by $B$, we obtain

$$
F^*\left(\theta^{(t+1)}\right) \leq F^*\left(\theta^{(t)}\right) + \left\langle\theta^{(t+1)} - \theta^{(t)}, \frac{d}{d\theta}F^*\left(\theta^{(t)}\right)\right\rangle + \frac{L_6}{2}\left\|\theta^{(t+1)} - \theta^{(t)}\right\|^2 \tag{84}
$$

where $F^*(\theta) = F(\theta, \phi^*(\theta))$ is the optimal outer level objective function. Then, the difference of the optimal outer level objective function $F^*\left(\theta^{(t+1)}\right) - F^*\left(\theta^{(t)}\right)$ is as follows.

$$
F^*\left(\theta^{(t+1)}\right) - F^*\left(\theta^{(t)}\right) \leq \left\langle\theta^{(t+1)} - \theta^{(t)}, \frac{d}{d\theta}F^*\left(\theta^{(t)}\right)\right\rangle + \frac{L_6}{2}\left\|\theta^{(t+1)} - \theta^{(t)}\right\|^2
$$

$$
= \left\langle\theta^{(t+1)} - \theta^{(t)}, \frac{d}{d\theta}F^*\left(\theta^{(t)}\right) - \mathbb{E}\left[\frac{\hat{d}}{d\theta}F\left(\theta^{(t)}, \hat{\phi}\left(\theta^{(t)}\right)\right)\right] - B_t\right\rangle
$$

$$
+ \left\langle\theta^{(t+1)} - \theta^{(t)}, \mathbb{E}\left[\frac{\hat{d}}{d\theta}F\left(\theta^{(t)}, \hat{\phi}\left(\theta^{(t)}\right)\right)\right] + B_t\right\rangle
$$

$$
+ \frac{L_6}{2}\left\|\theta^{(t+1)} - \theta^{(t)}\right\|^2 \tag{85}
$$

Because the meta-parameter of the Nash-GBML algorithm is updated as $\theta^{(t+1)} = \theta^{(t)} - \beta\hat{h}_\theta\left(\theta^{(t)}, \hat{\phi}\left(\theta^{(t)}\right)\right)$, that is the difference is $\theta^{(t+1)} - \theta^{(t)} = -\beta\hat{h}_\theta\left(\theta^{(t)}, \hat{\phi}\left(\theta^{(t)}\right)\right)$. Multiplying both sides by $\theta^{(t+1)} - \theta^{(t)}$ and simplifying, we get

$$
\left\langle\theta^{(t+1)} - \theta^{(t)}, \hat{h}_\theta\left(\theta^{(t)}, \hat{\phi}\left(\theta^{(t)}\right)\right)\right\rangle = -\frac{1}{\beta}\left\|\theta^{(t+1)} - \theta^{(t)}\right\|^2 \tag{86}
$$

By combining equations (85) and (86),

$$
\begin{aligned}
F^* \left(\theta^{(t+1)}\right) - F^* \left(\theta^{(t)}\right) \leq & \left\langle \theta^{(t+1)} - \theta^{(t)}, \frac{d}{d\theta} F^* \left(\theta^{(t)}\right) - \mathbb{E}\left[\frac{\hat{d}}{d\theta} F\left(\theta^{(t)}, \hat{\phi}\left(\theta^{(t)}\right)\right)\right] - B_t \right\rangle \\
& + \left\langle \theta^{(t+1)} - \theta^{(t)}, \mathbb{E}\left[\frac{\hat{d}}{d\theta} F\left(\theta^{(t)}, \hat{\phi}\left(\theta^{(t)}\right)\right)\right] + B_t - \hat{h}_\theta\left(\theta^{(t)}, \hat{\phi}\left(\theta^{(t)}\right)\right) \right\rangle \\
& - \frac{1}{\beta} \left\|\theta^{(t+1)} - \theta^{(t)}\right\|^2 + \frac{L_6}{2} \left\|\theta^{(t+1)} - \theta^{(t)}\right\|^2
\end{aligned} \tag{87}
$$

Because $\langle a, b \rangle \leq \frac{1}{2c} \|a\|^2 + \frac{c}{2} \|b\|^2$ for some constant $c$, the following equation holds under the definition of $\hat{h}_\theta\left(\theta^{(t)}, \hat{\phi}\left(\theta^{(t)}\right)\right)$.

$$
\begin{aligned}
F^* \left(\theta^{(t+1)}\right) - F^* \left(\theta^{(t)}\right) \leq & \frac{1}{2c_1} \left\|\frac{d}{d\theta} F^* \left(\theta^{(t)}\right) - \mathbb{E}\left[\frac{\hat{d}}{d\theta} F\left(\theta^{(t)}, \hat{\phi}\left(\theta^{(t)}\right)\right)\right] - B_t \right\|^2 \\
& + \frac{1}{2c_2} \left\|\mathbb{E}\left[\hat{h}_\theta\left(\theta^{(t)}, \hat{\phi}\left(\theta^{(t)}\right)\right)\right] - \hat{h}_\theta\left(\theta^{(t)}, \hat{\phi}\left(\theta^{(t)}\right)\right)\right\|^2 \\
& + \left(\frac{c_1 + c_2 + L_6}{2} - \frac{1}{\beta}\right) \left\|\theta^{(t+1)} - \theta^{(t)}\right\|^2
\end{aligned} \tag{88}
$$

Under Assumption D.4, the expectation of equation (88) is as follows.

$$
\begin{aligned}
\mathcal{L}\left(\theta^{(t+1)}\right) - \mathcal{L}\left(\theta^{(t)}\right) = & \mathbb{E}\left[F\left(\theta^{(t+1)}, \phi^*\left(\theta^{(t+1)}\right)\right)\right] - \mathbb{E}\left[F\left(\theta^{(t)}, \phi^*\left(\theta^{(t)}\right)\right)\right] \\
= & \mathbb{E}\left[F^*\left(\theta^{(t+1)}\right) - F^*\left(\theta^{(t)}\right)\right] \\
\leq & \frac{1}{2c_1} \mathbb{E}\left[\left\|\frac{d}{d\theta} F^*\left(\theta^{(t)}\right) - \mathbb{E}\left[\frac{\hat{d}}{d\theta} F\left(\theta^{(t)}, \hat{\phi}\left(\theta^{(t)}\right)\right)\right] - B_t\right\|^2\right] \\
& + \frac{\sigma_t^2}{2c_2} + \left(\frac{c_1 + c_2 + L_6}{2} - \frac{1}{\beta}\right) \mathbb{E}\left[\left\|\theta^{(t+1)} - \theta^{(t)}\right\|^2\right] \\
\leq & \frac{2}{c_1} \mathbb{E}\left[\left\|\frac{d}{d\theta} F^*\left(\theta^{(t)}\right) - \mathbb{E}\left[\frac{d}{d\theta} F^*\left(\theta^{(t)}\right)\right]\right\|^2\right] \\
& + \frac{2}{c_1} \mathbb{E}\left[\left\|\mathbb{E}\left[\frac{d}{d\theta} F^*\left(\theta^{(t)}\right)\right] - \mathbb{E}\left[\frac{\hat{d}}{d\theta} F\left(\theta^{(t)}, \hat{\phi}\left(\theta^{(t)}\right)\right)\right]\right\|^2\right] \\
& + \frac{b_t^2}{c_1} + \frac{\sigma_t^2}{2c_2} + \left(\frac{c_1 + c_2 + L_6}{2} - \frac{1}{\beta}\right) \mathbb{E}\left[\left\|\theta^{(t+1)} - \theta^{(t)}\right\|^2\right]
\end{aligned} \tag{89}
$$

Let $\frac{\hat{d}}{d\theta}F\left(\theta^{(t)}\right) = \frac{\hat{d}}{d\theta}F\left(\theta^{(t)}, \hat{\phi}\left(\theta^{(t)}\right)\right)$. By Lemma D.2 and Jensen's inequality, the first term and second term of equation (89) is as follows Under Assumption D.4.

$$\mathbb{E}\left[\left\|\frac{d}{d\theta}F^*\left(\theta^{(t)}\right) - \mathbb{E}\left[\frac{d}{d\theta}F^*\left(\theta^{(t)}\right)\right]\right\|^2\right] \leq \bar{\sigma}_t^2 \tag{90}$$

$$\mathbb{E}\left[\left\|\mathbb{E}\left[\frac{d}{d\theta}F^*\left(\theta^{(t)}\right)\right] - \mathbb{E}\left[\frac{\hat{d}}{d\theta}F\left(\theta^{(t)}\right)\right]\right\|^2\right] = \mathbb{E}\left[\left\|\mathbb{E}\left[\frac{d}{d\theta}F^*\left(\theta^{(t)}\right) - \frac{\hat{d}}{d\theta}F\left(\theta^{(t)}\right)\right]\right\|^2\right]$$

$$\leq \mathbb{E}\left[\mathbb{E}\left[\left\|\frac{d}{d\theta}F^*\left(\theta^{(t)}\right) - \frac{\hat{d}}{d\theta}F\left(\theta^{(t)}\right)\right\|^2\right]\right]$$

$$= \mathbb{E}\left[\left\|\frac{d}{d\theta}F^*\left(\theta^{(t)}\right) - \frac{\hat{d}}{d\theta}F\left(\theta^{(t)}\right)\right\|\right]^2$$

$$\leq C^2\mathbb{E}\left[\left\|\phi^*\left(\theta^{(t)}\right) - \hat{\phi}\left(\theta^{(t)}\right)\right\|\right]^2 \tag{91}$$

Combining equations (89), (90), and (91), the difference of optimal outer level objective function is expanded as follows.

$$\mathcal{L}\left(\theta^{(t+1)}\right) - \mathcal{L}\left(\theta^{(t)}\right) \leq \frac{2}{c_1}\mathbb{E}\left[\left\|\frac{d}{d\theta}F^*\left(\theta^{(t)}\right) - \mathbb{E}\left[\frac{d}{d\theta}F^*\left(\theta^{(t)}\right)\right]\right\|^2\right]$$

$$+ \frac{2}{c_1}\mathbb{E}\left[\left\|\mathbb{E}\left[\frac{d}{d\theta}F^*\left(\theta^{(t)}\right)\right] - \mathbb{E}\left[\frac{\hat{d}}{d\theta}F\left(\theta^{(t)}, \hat{\phi}\left(\theta^{(t)}\right)\right)\right]\right\|^2\right]$$

$$+ \frac{b_t^2}{c_1} + \frac{\sigma_t^2}{2c_2} + \left(\frac{c_1 + c_2 + L_6}{2} - \frac{1}{\beta}\right)\mathbb{E}\left[\left\|\theta^{(t+1)} - \theta^{(t)}\right\|^2\right]$$

$$\leq \left(\frac{c_1 + c_2 + L_6}{2} - \frac{1}{\beta}\right)\mathbb{E}\left[\left\|\theta^{(t+1)} - \theta^{(t)}\right\|^2\right]$$

$$+ \frac{2C^2}{c_1}\mathbb{E}\left[\left\|\phi^*\left(\theta^{(t)}\right) - \hat{\phi}\left(\theta^{(t)}\right)\right\|\right]^2 + \frac{2\bar{\sigma}_t^2 + b_t^2}{c_1} + \frac{\sigma_t^2}{2c_2} \tag{92}$$

where $C = L_1 + \frac{C_1 L_4 + C_2 L_2}{\mu_1 + \mu_2} + \frac{C_1 C_2 (L_3 + L_5)}{(\mu_1 + \mu_2)^2}$. Let $c_1 = c_2 = \frac{1}{2\beta}$. Then, the expectation of the difference of outer level objective function is simplified.

$$\mathcal{L}\left(\theta^{(t+1)}\right) - \mathcal{L}\left(\theta^{(t)}\right) \leq \left(\frac{c_1 + c_2 + L_6}{2} - \frac{1}{\beta}\right)\mathbb{E}\left[\left\|\theta^{(t+1)} - \theta^{(t)}\right\|^2\right]$$

$$+ \frac{2C^2}{c_1}\mathbb{E}\left[\left\|\phi^*\left(\theta^{(t)}\right) - \hat{\phi}\left(\theta^{(t)}\right)\right\|\right]^2 + \frac{2\bar{\sigma}_t^2 + b_t^2}{c_1} + \frac{\sigma_t^2}{2c_2}$$

$$\leq \left(\frac{L_6}{2} - \frac{1}{2\beta}\right)\mathbb{E}\left[\left\|\theta^{(t+1)} - \theta^{(t)}\right\|^2\right]$$

$$+ 4\beta C^2\mathbb{E}\left[\left\|\phi^*\left(\theta^{(t)}\right) - \hat{\phi}\left(\theta^{(t)}\right)\right\|\right]^2 + 4\beta\bar{\sigma}_t^2 + 2\beta b_t^2 + \beta\sigma_t^2 \tag{93}$$

Using the lemmas we proved earlier, we present that the Nash-GBML algorithm always converges. Moreover, we prove that the convergent point of the Nash-GBML algorithm is the Stackelberg equilibrium of the stochastic optimization problem described in equations (10) and (11). First, let's discuss the convergence of the Nash-GBML algorithm.

**Theorem D.7** *Let $(\theta^*, \phi^*(\theta^*))$ be a convergent point of the Nash-GBML algorithm, and $\mathcal{L}(\theta)$ be an expected optimal meta loss function, that is, $\mathcal{L}(\theta) = \mathbb{E}_{\mathcal{T}_i \sim p(\mathcal{T})}[F(\theta, \phi^*(\theta))]$. We denote $\delta$ as the convergence criterion of the inner level and $\bar{\delta}$ as the convergence criterion of the outer level.*

*That is, the inner level is converged when $\left\|\phi^*\left(\theta\right)-\hat{\phi}\right\| \le \delta$ and the outer level is converged when $\left\|\theta^{(t+1)}-\theta^{(t)}\right\| \le \bar{\delta}$. Under Assumption D.4, the following statements hold.*

- *The expected difference of the meta-parameter $\theta$ is bounded as follows.*

$$\mathbb{E}_{\mathcal{T}_i \sim p(\mathcal{T})}\left[\left\|\theta^{(t+1)}-\theta^{(t)}\right\|^2\right] \le 4\beta^2 C^2 \mathbb{E}\left[\left\|\hat{\phi}\left(\theta^{(t)}\right)-\phi^*\left(\theta^{(t)}\right)\right\|\right]^2$$

$$+\beta^2\sigma_t^2+2\beta^2 b_t^2+4\beta^2\left(\bar{C}_1+\frac{C_1 C_2}{\mu_1+\mu_2}\right)^2$$

$$\le 4\beta^2 C^2\delta^2+\beta^2\sigma_t^2+2\beta^2 b_t^2$$

$$+4\beta^2\left(\bar{C}_1+\frac{C_1 C_2}{\mu_1+\mu_2}\right)^2 \tag{94}$$

- *The expected difference of the optimal meta loss function $F^*\left(\theta\right)$ is bounded as follows.*

$$\mathcal{L}\left(\theta^{(t+1)}\right)-\mathcal{L}\left(\theta^{(t)}\right) \le 4\beta^2 C^2\left(\frac{L_6}{2}+\frac{1}{2\beta}\right)\mathbb{E}\left[\left\|\hat{\phi}\left(\theta^{(t)}\right)-\phi^*\left(\theta^{(t)}\right)\right\|\right]^2$$

$$+\left(\frac{L_6}{2}-\frac{1}{2\beta}\right)\left(\beta^2\sigma_t^2+2\beta^2 b_t^2+4\beta^2\left(\bar{C}_1+\frac{C_1 C_2}{\mu_1+\mu_2}\right)^2\right)$$

$$+4\beta\bar{\sigma}_t^2+2\beta b_t^2+\beta\sigma_t^2$$

$$\le \left(\frac{L_6}{2}-\frac{1}{2\beta}\right)\left(\beta^2\sigma_t^2+2\beta^2 b_t^2+4\beta^2\left(\bar{C}_1+\frac{C_1 C_2}{\mu_1+\mu_2}\right)^2\right)$$

$$+4\beta\bar{\sigma}_t^2+2\beta b_t^2+\beta\sigma_t^2+4\beta^2 C^2\delta^2\left(\frac{L_6}{2}+\frac{1}{2\beta}\right) \tag{95}$$

- *We denote the convergence speed of each non-increasing sequence $\{b_k\}_{k\ge 0}$, $\{\sigma_k\}_{k\ge 0}$, and $\{\bar{\sigma}_k\}_{k\ge 0}$, which is defined in Assumption D.4, as $O\left(k_b\right)$, $O\left(k_\sigma\right)$, and $O\left(\bar{k}_{\bar{\sigma}}\right)$, respectively. After we choose the step size*

$$\beta \le \frac{\bar{\delta}}{\sqrt{4C^2\delta^2+4\left(\bar{C}_1+\frac{C_1 C_2}{\mu_1+\mu_2}\right)^2}} \tag{96}$$

*, the iteration complexity of the Nash-GBML algorithm's outer level is $O\left(\max\left\{k_b^2, k_\sigma^2, k_{\bar{\sigma}}^2\right\}\right)$ and the expected error of the optimal meta loss function of the convergent point is*

$$\mathcal{L}\left(\theta^*\right)-\mathcal{L}\left(\theta^{(t)}\right) \le \frac{L_6}{2}\bar{\delta}^2+\frac{C^2\delta^2-\left(\bar{C}_1+\frac{C_1 C_2}{\mu_1+\mu_2}\right)^2}{\sqrt{C^2\delta^2+\left(\bar{C}_1+\frac{C_1 C_2}{\mu_1+\mu_2}\right)^2}}\bar{\delta} \tag{97}$$

*where $C=L_1+\frac{C_1 L_4+C_2 L_2}{\mu_1+\mu_2}+\frac{C_1 C_2 (L_3+L_5)}{(\mu_1+\mu_2)^2}$.*

*Proof.* The gradient update procedure of the meta-parameter for the Nash-GBML algorithm is $\theta^{(t+1)}=\theta^{(t)}-\beta\hat{h}_\theta\left(\theta^{(t)},\hat{\phi}\left(\theta^{(t)}\right)\right)$. Thus, the expectation of the meta-parameter is as follows by Lemma D.5.

$$\left\|\theta^{(t+1)}-\theta^{(t+1)}\right\|=\beta\left\|\hat{h}_\theta\left(\theta^{(t+1)},\hat{\phi}\left(\theta^{(t+1)}\right)\right)\right\| \tag{98}$$

$$\mathbb{E}\left[\left\|\theta^{(t+1)}-\theta^{(t+1)}\right\|^2\right]=\beta^2\mathbb{E}\left[\left\|\hat{h}_\theta\left(\theta^{(t+1)},\hat{\phi}\left(\theta^{(t+1)}\right)\right)\right\|^2\right]$$

$$\le 4\beta^2 C^2\mathbb{E}\left[\left\|\hat{\phi}\left(\theta^{(t+1)}\right)-\phi^*\left(\theta^{(t+1)}\right)\right\|\right]^2$$

$$+\beta^2\sigma_t^2+2\beta^2 b_t^2+4\beta^2\left(\bar{C}_1+\frac{C_1 C_2}{\mu_1+\mu_2}\right)^2 \tag{99}$$

where $C = L_1 + \frac{C_1 L_4 + C_2 L_2}{\mu_1 + \mu_2} + \frac{C_1 C_2 (L_3 + L_5)}{(\mu_1 + \mu_2)^2}$.

By Theorem C.3, $\delta$-accurate estimation of the optimal joint task-specific parameter is computed with $O\left(\kappa \log\left(D/\delta\right)\right)$ number of iterations Under Assumption B.1.

$$\left\| \hat{\phi}\left(\theta^{(t)}\right) - \phi^*\left(\theta^{(t)}\right) \right\| \leq \delta, \forall k \tag{100}$$

Now we derive the expected difference of the optimal meta-parameter and its loss function by equations (99) and (82).

$$\mathbb{E}\left[\left\| \theta^{(t+1)} - \theta^{(t)} \right\|^2\right] \leq 4\beta^2 C^2 \mathbb{E}\left[\left\| \hat{\phi}\left(\theta^{(t)}\right) - \phi^*\left(\theta^{(t)}\right) \right\|\right]^2$$

$$+ \beta^2 \sigma_t^2 + 2\beta^2 b_t^2 + 4\beta^2 \left(\bar{C}_1 + \frac{C_1 C_2}{\mu_1 + \mu_2}\right)^2$$

$$\leq 4\beta^2 C^2 \delta^2 + \beta^2 \sigma_t^2 + 2\beta^2 b_t^2 + 4\beta^2 \left(\bar{C}_1 + \frac{C_1 C_2}{\mu_1 + \mu_2}\right)^2 \tag{101}$$

$$\mathcal{L}\left(\theta^{(t+1)}\right) - \mathcal{L}\left(\theta^{(t)}\right) \leq \left(\frac{L_6}{2} - \frac{1}{2\beta}\right) \mathbb{E}\left[\left\| \theta^{(t+1)} - \theta^{(t)} \right\|^2\right]$$

$$+ 4\beta C^2 \mathbb{E}\left[\left\| \phi^*\left(\theta^{(t)}\right) - \hat{\phi}\left(\theta^{(t)}\right) \right\|\right]^2 + 4\beta \bar{\sigma}_t^2 + 2\beta b_t^2 + \beta \sigma_t^2$$

$$\leq 4\beta^2 C^2 \left(\frac{L_6}{2} + \frac{1}{2\beta}\right) \mathbb{E}\left[\left\| \hat{\phi}\left(\theta^{(t)}\right) - \phi^*\left(\theta^{(t)}\right) \right\|\right]^2$$

$$+ \left(\frac{L_6}{2} - \frac{1}{2\beta}\right)\left(\beta^2 \sigma_t^2 + 2\beta^2 b_t^2 + 4\beta^2 \left(\bar{C}_1 + \frac{C_1 C_2}{\mu_1 + \mu_2}\right)^2\right)$$

$$+ 4\beta \bar{\sigma}_t^2 + 2\beta b_t^2 + \beta \sigma_t^2$$

$$\leq \left(\frac{L_6}{2} - \frac{1}{2\beta}\right)\left(\beta^2 \sigma_t^2 + 2\beta^2 b_t^2 + 4\beta^2 \left(\bar{C}_1 + \frac{C_1 C_2}{\mu_1 + \mu_2}\right)^2\right)$$

$$+ 4\beta \bar{\sigma}_t^2 + 2\beta b_t^2 + \beta \sigma_t^2 + 4\beta^2 C^2 \delta^2 \left(\frac{L_6}{2} + \frac{1}{2\beta}\right) \tag{102}$$

The convergence of the Nash-GBML algorithm is guaranteed while the meta-parameter satisfies the convergence criterion of the outer level. Thus, the following equation holds by equation (101)

$$4\beta^2 C^2 \delta^2 + \beta^2 \sigma_t^2 + 2\beta^2 b_t^2 + 4\beta^2 \left(\bar{C}_1 + \frac{C_1 C_2}{\mu_1 + \mu_2}\right)^2 \leq \bar{\delta}^2$$

$$\beta^2 \left(4 C^2 \delta^2 + \sigma_t^2 + 2 b_t^2 + 4\left(\bar{C}_1 + \frac{C_1 C_2}{\mu_1 + \mu_2}\right)^2\right) \leq \bar{\delta}^2 \tag{103}$$

Because the convergence speed of $\sigma_t^2 + 2b_t^2$ is $O\left(\max\left\{k_b^2, k_\sigma^2\right\}\right)$, step size $\beta$ should be less than

$$\beta \leq \frac{\bar{\delta}}{\sqrt{4 C^2 \delta^2 + 4\left(\bar{C}_1 + \frac{C_1 C_2}{\mu_1 + \mu_2}\right)^2}} \tag{104}$$

and the difference of expected optimal meta loss $\mathcal{L}\left(\theta^{(t+1)}\right) - \mathcal{L}\left(\theta^{(t)}\right)$ holds by equation (102)

$$
\mathcal{L}\left(\theta^{(t+1)}\right) - \mathcal{L}\left(\theta^{(t)}\right) \leq \left(\frac{L_6}{2} - \frac{1}{2\beta}\right)\left(\beta^2\sigma_t^2 + 2\beta^2 b_t^2 + 4\beta^2\left(\bar{C}_1 + \frac{C_1 C_2}{\mu_1 + \mu_2}\right)^2\right)
$$

$$
+ 4\beta\bar{\sigma}_t^2 + 2\beta b_t^2 + \beta\sigma_t^2 + 4\beta^2 C^2\delta^2\left(\frac{L_6}{2} + \frac{1}{2\beta}\right)
$$

$$
\leq 4\beta^2\left(\frac{L_6}{2} - \frac{1}{2\beta}\right)\left(\bar{C}_1 + \frac{C_1 C_2}{\mu_1 + \mu_2}\right)^2 + 4\beta^2 C^2\delta^2\left(\frac{L_6}{2} + \frac{1}{2\beta}\right)
$$

$$
\leq \frac{L_6\bar{\delta}^2}{2} - \frac{\bar{\delta}^2}{2\beta} + 4\beta C^2\delta^2
$$

$$
\leq \frac{L_6}{2}\bar{\delta}^2 + \frac{C^2\delta^2 - \left(\bar{C}_1 + \frac{C_1 C_2}{\mu_1 + \mu_2}\right)^2}{\sqrt{C^2\delta^2 + \left(\bar{C}_1 + \frac{C_1 C_2}{\mu_1 + \mu_2}\right)^2}}\bar{\delta} \tag{105}
$$

at the convergent point $(\theta^*, \phi^*(\theta^*))$ with convergent speed $O\left(\max\left\{k_b^2, k_\sigma^2, k_{\bar{\sigma}}^2\right\}\right)$. That is, the error of the expected meta loss at the convergent point is less than

$$
\frac{L_6}{2}\bar{\delta}^2 + \frac{C^2\delta^2 - \left(\bar{C}_1 + \frac{C_1 C_2}{\mu_1 + \mu_2}\right)^2}{\sqrt{C^2\delta^2 + \left(\bar{C}_1 + \frac{C_1 C_2}{\mu_1 + \mu_2}\right)^2}}\bar{\delta} \tag{106}
$$

Next, we discuss the convergent point of the Nash-GBML algorithm is a Stackelberg equilibrium of the target stochastic optimization problem of the Nash-GBML algorithm described in equations (10) and (11).

**Lemma D.8** *The optimal expected meta loss function $\mathcal{L}(\theta)$ is always equal to the task $i$'s expected loss function $\mathbb{E}_{\mathcal{T}_i \sim p(\mathcal{T})}[\mathcal{L}_{\mathcal{T}_i}(\theta)]$.*

*Proof.* Because the sampling task is done through replacement sampling, the following holds.

$$
\mathcal{L}(\theta) = \mathbb{E}_{\mathcal{T}_i \sim p(\mathcal{T})}[F(\theta, \phi^*(\theta))]
$$

$$
= \frac{1}{B}\sum_{i=1}^{B}\mathbb{E}_{\mathcal{T}_i \sim p(\mathcal{T})}[\mathcal{L}_{\mathcal{T}_i}(\theta, \phi^*(\theta))]
$$

$$
= \mathbb{E}_{\mathcal{T}_i \sim p(\mathcal{T})}[\mathcal{L}_{\mathcal{T}_i}(\theta, \phi^*(\theta))] \tag{107}
$$

**Theorem D.9** *Let $(\theta^*, \phi^*(\theta^*))$ be an optimal solution of the following stochastic optimization problem which is the target problem of the Nash-GBML algorithm.*

$$
\theta^* = \arg\min_{\theta \in \mathbb{R}^d}\mathbb{E}_{\mathcal{T}_i \sim p(\mathcal{T})}[\mathcal{L}_{\mathcal{T}_i}(\theta, \phi_i^*(\theta))] \tag{108}
$$

$$
\phi_i^*(\theta) = \arg\min_{\phi_i \in \Omega_i} f_i(\phi_i, \phi_{-i}^*(\theta), \theta) \tag{109}
$$

*We denote the expected meta loss function of the stochastic optimization problem $\mathbb{E}_{\mathcal{T}_i \sim p(\mathcal{T})}[\mathcal{L}_{\mathcal{T}_i}(\theta, \phi^*(\theta))]$ as $\mathbb{E}\left[\mathcal{L}_{\mathcal{T}_i}^*(\theta)\right]$. Let $\delta$ and $\bar{\delta}$ be the convergence criterion of the inner level and the outer level, respectively. Then, under Assumption D.4, the Nash-GBML algorithm with step size $\beta \leq \frac{\bar{\delta}}{\sqrt{4C^2\delta^2 + 4\left(\bar{C}_1 + \frac{C_1 C_2}{\mu_1 + \mu_2}\right)^2}}$ compute the optimal solution of the stochastic optimization problem described in equations (108) and (109) with the convergence speed $O\left(\max\left\{k_b^2, k_\sigma^2, k_{\bar{\sigma}}^2\right\}\right)$ and error*

$$
\mathbb{E}\left[\mathcal{L}_{\mathcal{T}_i}^*(\theta^*)\right] - \mathbb{E}\left[\mathcal{L}_{\mathcal{T}_i}^*\left(\theta^{(t)}\right)\right] \leq \frac{L_6}{2}\bar{\delta}^2 + \frac{C^2\delta^2 - \left(\bar{C}_1 + \frac{C_1 C_2}{\mu_1 + \mu_2}\right)^2}{\sqrt{C^2\delta^2 + \left(\bar{C}_1 + \frac{C_1 C_2}{\mu_1 + \mu_2}\right)^2}}\bar{\delta} \tag{110}
$$

where $C = L_1 + \frac{C_1 L_4 + C_2 L_2}{\mu_1 + \mu_2} + \frac{C_1 C_2 (L_3 + L_5)}{(\mu_1 + \mu_2)^2}$. *That is, the convergent point of the Nash-GBML algorithm is also the general Stackelberg equilibrium of the SLMF game described in equations (108) and (109).*

*Proof.* By Theorem D.7 and Lemma D.8, the difference of the expected meta loss function of the stochastic problem at the convergent point of the Nash-GBML algorithm is converged with convergent speed $O\left(\max\left\{k_b^2, k_\sigma^2, k_{\bar{\sigma}}^2\right\}\right)$.

$$\mathbb{E}\left[\mathcal{L}_{\mathcal{T}_i}^*\left(\theta^{(t+1)}\right)\right] - \mathbb{E}\left[\mathcal{L}_{\mathcal{T}_i}^*\left(\theta^{(t)}\right)\right] = \boldsymbol{\mathcal{L}}\left(\theta^{(t+1)}\right) - \boldsymbol{\mathcal{L}}\left(\theta^{(t)}\right)$$

$$\leq \frac{L_6}{2}\bar{\delta}^2 + \frac{C^2\delta^2 - \left(\bar{C}_1 + \frac{C_1 C_2}{\mu_1 + \mu_2}\right)^2}{\sqrt{C^2\delta^2 + \left(\bar{C}_1 + \frac{C_1 C_2}{\mu_1 + \mu_2}\right)^2}}\bar{\delta} \qquad (111)$$

where $C = L_1 + \frac{C_1 L_4 + C_2 L_2}{\mu_1 + \mu_2} + \frac{C_1 C_2 (L_3 + L_5)}{(\mu_1 + \mu_2)^2}$. That is, the error of the expected meta loss function $\mathbb{E}\left[\mathcal{L}_{\mathcal{T}_i}^*\left(\theta\right)\right]$ at the convergent point of the Nash-GBML algorithm is less than

$$\frac{L_6}{2}\bar{\delta}^2 + \frac{C^2\delta^2 - \left(\bar{C}_1 + \frac{C_1 C_2}{\mu_1 + \mu_2}\right)^2}{\sqrt{C^2\delta^2 + \left(\bar{C}_1 + \frac{C_1 C_2}{\mu_1 + \mu_2}\right)^2}}\bar{\delta} \qquad (112)$$

Because the difference of the expected meta loss function $\mathbb{E}\left[\mathcal{L}_{\mathcal{T}_i}^*\left(\theta\right)\right]$ is converged in equation (111), the convergent point of the Nash-GBML algorithm is also the optimal solution of the stochastic optimization problem described in equations (108) and (109). That is,

$$\mathbb{E}\left[\mathcal{L}_{\mathcal{T}_i}^*\left(\theta^*\right)\right] - \mathbb{E}\left[\mathcal{L}_{\mathcal{T}_i}^*\left(\theta^{(t)}\right)\right] \leq \frac{L_6}{2}\bar{\delta}^2 + \frac{C^2\delta^2 - \left(\bar{C}_1 + \frac{C_1 C_2}{\mu_1 + \mu_2}\right)^2}{\sqrt{C^2\delta^2 + \left(\bar{C}_1 + \frac{C_1 C_2}{\mu_1 + \mu_2}\right)^2}}\bar{\delta} \qquad (113)$$

So far, we show that the Nash-GBML algorithm converges, and its convergent point is the Stackelberg equilibrium of the stochastic SLMF game, which is the target problem of Nash-GBML algorithm. Now, we prove that the Nash-GBML algorithm always converges to the same point regardless of the initial meta-parameter or initial task-specific parameter, and irrespective of the order in which task-specific parameters are updated in the inner level.

**Theorem D.10** *Under Assumption D.4, the Nash-GBML algorithm converges to the same optimal solution of the stochastic optimization problem described in equations (108) and (109) regardless of the order of the task-specific parameters' gradient update in the inner level. Moreover, the Nash-GBML algorithm converges to the optimal solution of the stochastic optimization problem described in equations (108) and (109) regardless of the initial meta-parameter and initial task-specific parameters.*

*Proof.* Let $G\left(\theta\right)$ be the $N$-player normal-form game modeling an inner level of the Nash-GBML algorithm. Because there is the unique convergent point of $G\left(\theta\right)$ by Lemma B.3, the Nash-GBML algorithm converges to the same convergent point regardless of the order of the task-specific parameters' gradient update.

$G\left(\theta\right)$ has the unique Nash equilibrium under Assumption B.1. Thus, there is the unique Nash equilibrium $\phi^*\left(\theta\right)$ of equation (109) under Assumption D.4. Because the optimal task-specific loss function is convex on $\theta$, $\mathbb{E}_{\mathcal{T}_i \sim p(\mathcal{T})}\left[\mathcal{L}_{\mathcal{T}_i}\left(\theta, \phi^*\left(\theta\right)\right)\right]$ is convex on $\theta$. Therefore, there is the unique optimal solution $\left(\theta^*, \phi^*\left(\theta^*\right)\right)$ of the stochastic optimization problem described in equations (108) and (109). That is, the Nash-GBML algorithm converges to the optimal solution of the stochastic optimization problem regardless of the initial meta-parameter and initial task-specific parameters.

## E PENALTY TERM

The penalty term $p_C$ is designed to minimize the distance between meta-parameter $\theta$ and the center of task-specific parameters $\frac{1}{N} \sum_{i=1}^{N} \phi_i$. To scale it to the task-specific parameters in the batch, the distance is represented as follows approximately:

$$
\begin{aligned}
\left\| \theta - \frac{1}{N} \sum_{k=i}^{N} \phi_k \right\|_2^2 &= \left\| \frac{1}{N} \sum_{k=i}^{N} (\theta - \phi_k) \right\|_2^2 \\
&\approx \left\| \frac{B}{N} \frac{1}{B} \sum_{k=i}^{B} (\theta - \phi_k) \right\|_2^2 \\
&= \left( \frac{B}{N} \right)^2 \left\| \frac{1}{B} \sum_{k=i}^{B} (\theta - \phi_k) \right\|_2^2 \\
&= \left( \frac{B}{N} \right)^2 \left\| \theta - \frac{1}{B} \sum_{k=i}^{B} \phi_k \right\|_2^2
\end{aligned}
\tag{114}
$$

Because $\theta - \phi_k$ is proportional to the inner learning rate in Nash-GBML with the one-step gradient update. Thus, we divide equation (114) by $\alpha^2$. That is, the proportional constant is defined as $\left( \frac{B}{\alpha N} \right)^2$. We set $\alpha = 0.05$ instead of the inner learning rate for Meta-SGD and set $\theta = \phi_0$ for CAVIA.

# F ALGORITHM DETAIL AND ADDITIONAL EXPERIMENT

For all models, we perform experiments with Intel(R) Core(TM) i7-9700F CPU @ 3.00GHz, 16.0GB RAM, and a single NVIDIA GeForce RTX 2060 GPU.

Every algorithm for sinusoid regression (Collins et al., 2020) uses the same fully connected network architecture described in (Finn et al., 2017). We evaluate models with three-step gradient update where a fixed learning rate $(\alpha, \beta) = (0.001, 0.001)$ and train for 70000 iterations with a meta batch of 25 tasks. We set the task-probability learning rate as $0.00001$ for TR-MAML and use 4 context parameters for CAVIA.

For image completion task (Garnelo et al., 2018), we use the same fully connected network architecture described in (Zintgraf et al., 2019). We use 128 context parameters for CAVIA. We evaluate models with five-step gradient update where a fixed learning rate $(\alpha, \beta) = (0.001, 0.1)$ and train for 50000 iterations with a meta batch of 25 tasks. We use 128 context parameters for CAVIA.

For MiniImageNet task (Ravi & Larochelle, 2017), we use the same convolutional network architecture described in (Finn et al., 2017). The number in parentheses after CAVIA indicates the number of filters. We use 100 context parameters for CAVIA. Every models are trained using five-step gradient update and train for 60000 iterations with a meta batch of 2 tasks. We set to $(\alpha, \beta) = (0.0001, 0.01)$ for MAML, Meta-SGD, and set to $(\alpha, \beta) = (0.001, 1)$ for CAVIA(32), CAVIA(128).

## F.1 ROBUSTNESS OF THE HYPERPARAMETER

Obviously, if the weight of the penalty term is too small, Nash-GBML is similar to GBML, and if it is too large, Nash-GBML does not work. In this section, we demonstrate how robust the performance of the centroid penalty term $p_C(w)$ and the robust penalty term $p_R(w, r)$ is with respect to the weight $w$.

In Table 5 and 6, we present the quantitative results of Nash-GBML for different hyparparameters for sinusoid regression. We evaluate the robustness of the first penalty term $p_C(w)$ where $w \in \{10^x | x \in \mathbb{Z}, -6 \le x \le -1\}$, and the second penalty term $p_R(w, r)$ where $w \in \{10^x | x \in \mathbb{Z}, -5 \le x \le -2\}$ and $r \in \{1, 2\}$. We validate that Nash-GBML outperforms GBML in most settings.

Table 5: MSE for MAML on 5-shot 3-step sinusoid regression with $95\%$ confidence intervals over 5 random trials depending on the hyperparameters. Red color means the better performance than MAML, and the bold means the best value for each penalty terms.

| Algorithm | Mean | Worst | Std. Dev. |
|---|---|---|---|
| MAML (Finn et al., 2017) | $0.59 \pm 0.01$ | $3.18 \pm 0.86$ | $0.57 \pm 0.04$ |
| MAML $+p_C(0.1)$ | $\mathbf{0.54 \pm 0.01}$ | $3.00 \pm 0.79$ | $0.55 \pm 0.03$ |
| MAML $+p_C(0.01)$ | $0.63 \pm 0.02$ | $3.29 \pm 0.83$ | $0.62 \pm 0.03$ |
| MAML $+p_C(0.001)$ | $0.64 \pm 0.02$ | $3.38 \pm 0.85$ | $0.61 \pm 0.04$ |
| MAML $+p_C(0.0001)$ | $0.55 \pm 0.02$ | $3.56 \pm 0.75$ | $0.73 \pm 0.04$ |
| MAML $+p_C(0.00001)$ | $0.54 \pm 0.01$ | $2.88 \pm 0.62$ | $0.52 \pm 0.03$ |
| MAML $+p_C(0.000001)$ | $0.54 \pm 0.02$ | $2.71 \pm 0.58$ | $0.49 \pm 0.03$ |
| MAML $+p_R(0.01, 1)$ | $0.51 \pm 0.02$ | $2.98 \pm 0.77$ | $0.51 \pm 0.03$ |
| MAML $+p_R(0.001, 1)$ | $0.58 \pm 0.02$ | $3.06 \pm 0.28$ | $0.56 \pm 0.03$ |
| MAML $+p_R(0.0001, 1)$ | $0.56 \pm 0.02$ | $3.06 \pm 0.78$ | $0.54 \pm 0.04$ |
| MAML $+p_R(0.00001, 1)$ | $0.53 \pm 0.01$ | $2.83 \pm 0.47$ | $0.52 \pm 0.03$ |
| MAML $+p_R(0.01, 2)$ | $0.56 \pm 0.01$ | $\mathbf{2.81 \pm 0.59}$ | $0.51 \pm 0.02$ |
| MAML $+p_R(0.001, 2)$ | $0.55 \pm 0.01$ | $2.90 \pm 0.84$ | $0.53 \pm 0.02$ |
| MAML $+p_R(0.0001, 2)$ | $0.51 \pm 0.01$ | $2.81 \pm 0.81$ | $0.50 \pm 0.03$ |
| MAML $+p_R(0.00001, 2)$ | $0.58 \pm 0.02$ | $3.18 \pm 0.86$ | $0.57 \pm 0.04$ |

Table 6: MSE for Meta-SGD, CAVIA, and TR-MAML on 5-shot 3-step sinusoid regression with 95% confidence intervals over 5 random trials depending on the hyperparameters. Red color means the better performance than GBML, and the bold means the best value for each penalty terms.

| Algorithm | Mean | Worst | Std. Dev. |
|---|---|---|---|
| Meta-SGD (Li et al., 2017) | $0.19 \pm 0.01$ | $1.48 \pm 0.68$ | $0.18 \pm 0.03$ |
| Meta-SGD $+p_C$ (0.1) | $\mathbf{0.14 \pm 0.00}$ | $1.32 \pm 0.67$ | $0.17 \pm 0.02$ |
| Meta-SGD $+p_C$ (0.01) | $0.28 \pm 0.01$ | $1.78 \pm 0.52$ | $0.28 \pm 0.02$ |
| Meta-SGD $+p_C$ (0.001) | $0.22 \pm 0.01$ | $1.85 \pm 0.50$ | $0.24 \pm 0.02$ |
| Meta-SGD $+p_C$ (0.0001) | $0.22 \pm 0.01$ | $1.52 \pm 0.58$ | $0.23 \pm 0.02$ |
| Meta-SGD $+p_C$ (0.00001) | $0.20 \pm 0.01$ | $1.49 \pm 0.49$ | $0.22 \pm 0.02$ |
| Meta-SGD $+p_C$ (0.000001) | $0.18 \pm 0.01$ | $1.45 \pm 0.64$ | $0.20 \pm 0.02$ |
| Meta-SGD $+p_R$ (0.01, 1) | $0.19 \pm 0.00$ | $1.60 \pm 0.34$ | $0.21 \pm 0.01$ |
| Meta-SGD $+p_R$ (0.001, 1) | $0.17 \pm 0.01$ | $1.32 \pm 0.58$ | $0.17 \pm 0.02$ |
| Meta-SGD $+p_R$ (0.0001, 1) | $0.19 \pm 0.00$ | $1.44 \pm 0.52$ | $0.20 \pm 0.01$ |
| Meta-SGD $+p_R$ (0.00001, 1) | $0.14 \pm 0.00$ | $\mathbf{1.07 \pm 0.37}$ | $0.16 \pm 0.01$ |
| Meta-SGD $+p_R$ (0.01, 2) | $0.17 \pm 0.00$ | $1.30 \pm 0.37$ | $0.18 \pm 0.01$ |
| Meta-SGD $+p_R$ (0.001, 2) | $0.14 \pm 0.00$ | $1.30 \pm 0.39$ | $0.16 \pm 0.01$ |
| Meta-SGD $+p_R$ (0.0001, 2) | $0.24 \pm 0.02$ | $3.42 \pm 3.67$ | $0.31 \pm 0.01$ |
| Meta-SGD $+p_R$ (0.00001, 2) | $0.15 \pm 0.01$ | $1.21 \pm 0.67$ | $0.17 \pm 0.02$ |
| CAVIA (Zintgraf et al., 2019) | $0.14 \pm 0.01$ | $1.29 \pm 0.62$ | $0.14 \pm 0.02$ |
| CAVIA $+p_C$ (0.1) | $0.15 \pm 0.01$ | $1.50 \pm 1.11$ | $0.16 \pm 0.04$ |
| CAVIA $+p_C$ (0.01) | $0.15 \pm 0.01$ | $1.21 \pm 0.69$ | $0.16 \pm 0.04$ |
| CAVIA $+p_C$ (0.001) | $\mathbf{0.13 \pm 0.00}$ | $1.15 \pm 0.59$ | $0.14 \pm 0.02$ |
| CAVIA $+p_C$ (0.0001) | $0.13 \pm 0.01$ | $1.20 \pm 0.75$ | $0.15 \pm 0.03$ |
| CAVIA $+p_C$ (0.00001) | $0.15 \pm 0.01$ | $1.55 \pm 1.10$ | $0.16 \pm 0.03$ |
| CAVIA $+p_C$ (0.000001) | $0.14 \pm 0.01$ | $1.19 \pm 0.55$ | $0.15 \pm 0.03$ |
| CAVIA $+p_R$ (0.01, 1) | $0.25 \pm 0.01$ | $1.98 \pm 0.71$ | $0.25 \pm 0.01$ |
| CAVIA $+p_R$ (0.001, 1) | $0.16 \pm 0.01$ | $1.39 \pm 0.69$ | $0.18 \pm 0.02$ |
| CAVIA $+p_R$ (0.0001, 1) | $0.22 \pm 0.01$ | $1.64 \pm 0.72$ | $0.21 \pm 0.02$ |
| CAVIA $+p_R$ (0.00001, 1) | $0.12 \pm 0.01$ | $1.26 \pm 0.96$ | $0.15 \pm 0.04$ |
| CAVIA $+p_R$ (0.01, 2) | $0.26 \pm 0.02$ | $1.76 \pm 0.15$ | $0.28 \pm 0.03$ |
| CAVIA $+p_R$ (0.001, 2) | $0.13 \pm 0.01$ | $1.28 \pm 0.36$ | $0.15 \pm 0.02$ |
| CAVIA $+p_R$ (0.0001, 2) | $0.14 \pm 0.00$ | $1.37 \pm 0.64$ | $0.15 \pm 0.02$ |
| CAVIA $+p_R$ (0.00001, 2) | $0.13 \pm 0.01$ | $\mathbf{1.18 \pm 0.46}$ | $0.15 \pm 0.02$ |
| TR-MAML (Collins et al., 2020) | $0.62 \pm 0.02$ | $2.35 \pm 0.46$ | $0.29 \pm 0.02$ |
| TR-MAML $+p_C$ (0.1) | $0.62 \pm 0.02$ | $2.77 \pm 1.32$ | $0.32 \pm 0.04$ |
| TR-MAML $+p_C$ (0.01) | $\mathbf{0.51 \pm 0.02}$ | $2.33 \pm 1.23$ | $0.28 \pm 0.04$ |
| TR-MAML $+p_C$ (0.001) | $0.54 \pm 0.02$ | $2.31 \pm 0.87$ | $0.28 \pm 0.03$ |
| TR-MAML $+p_C$ (0.0001) | $0.54 \pm 0.02$ | $2.38 \pm 0.95$ | $0.28 \pm 0.03$ |
| TR-MAML $+p_C$ (0.00001) | $0.55 \pm 0.02$ | $2.23 \pm 0.53$ | $0.28 \pm 0.03$ |
| TR-MAML $+p_C$ (0.000001) | $0.55 \pm 0.02$ | $2.30 \pm 0.55$ | $0.29 \pm 0.03$ |
| TR-MAML $+p_R$ (0.01, 1) | $0.52 \pm 0.01$ | $\mathbf{2.20 \pm 1.12}$ | $0.26 \pm 0.04$ |
| TR-MAML $+p_R$ (0.001, 1) | $0.60 \pm 0.02$ | $2.32 \pm 1.03$ | $0.28 \pm 0.03$ |
| TR-MAML $+p_R$ (0.0001, 1) | $0.63 \pm 0.02$ | $2.58 \pm 1.28$ | $0.33 \pm 0.04$ |
| TR-MAML $+p_R$ (0.00001, 1) | $0.63 \pm 0.02$ | $2.40 \pm 1.03$ | $0.30 \pm 0.04$ |
| TR-MAML $+p_R$ (0.01, 2) | $0.51 \pm 0.02$ | $2.42 \pm 1.18$ | $0.28 \pm 0.03$ |
| TR-MAML $+p_R$ (0.001, 2) | $0.61 \pm 0.02$ | $2.45 \pm 0.54$ | $0.31 \pm 0.04$ |
| TR-MAML $+p_R$ (0.0001, 2) | $0.54 \pm 0.02$ | $2.25 \pm 1.14$ | $0.28 \pm 0.04$ |
| TR-MAML $+p_R$ (0.00001, 2) | $0.52 \pm 0.02$ | $2.56 \pm 1.12$ | $0.31 \pm 0.05$ |

## F.2 ROBUSTNESS OF THE BATCH SIZE

We validate the performance of Nash-GBML depending on the batch size $B \in \{5, 10, 15, 20, 25\}$. In Figure F.2, F.2, F.2, and F.2, we show that Nash-GBML outperforms the traditional GBML in most settings. In particular, Nash-GBML algorithms consistently outperform GBML algorithms when the batch-size is large enough.

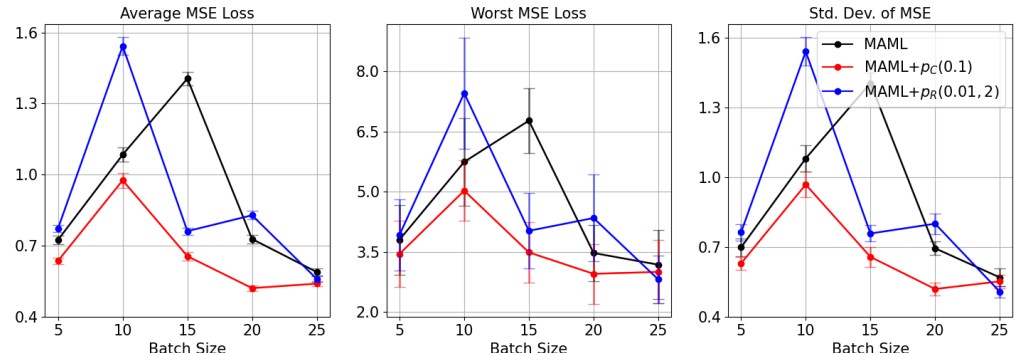

Figure 3: Test MSE statistics with 95% confidence intervals depends on the batch size for 5-shot sinusoid regression. The leftmost plot shows the average MSE loss and the center plot shows the worst MSE loss for MAML and the Nash-GBML that combines MAML with penalty terms. The statistics are empirical averages over 5000 samples.

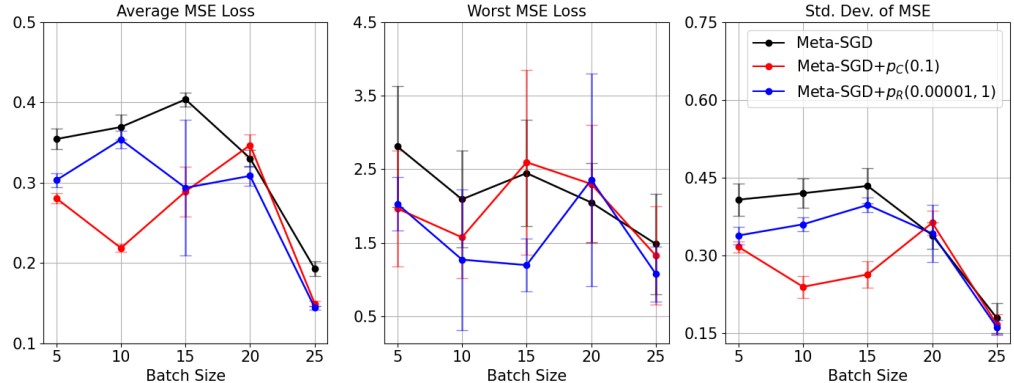

Figure 4: Test MSE statistics with 95% confidence intervals depends on the batch size for 5-shot sinusoid regression. The left plot shows the average MSE loss and the right plot shows the worst MSE loss for Meta-SGD and the Nash-GBML that combines Meta-SGD with penalty terms. The statistics are empirical averages over 5000 samples.

### F.3    ROBUSTNESS OF THE STEP SIZE

We validate the performance of Nash-GBML depending on the batch size $B \in \{5, 10, 15, 20, 25\}$. In Figure F.3, F.3, F.3, and F.3, we show that Nash-GBML outperforms the traditional GBML in most settings. Note that, you should use a small weight when the step size is large.

### F.4    CONVERGENCE TRAJECTORY

The convergence trajectory of Meta-SGD, CAVIA, TR-MAML, and its Nash-GBML is demonstrated in Figure F.4, F.4, and F.4.

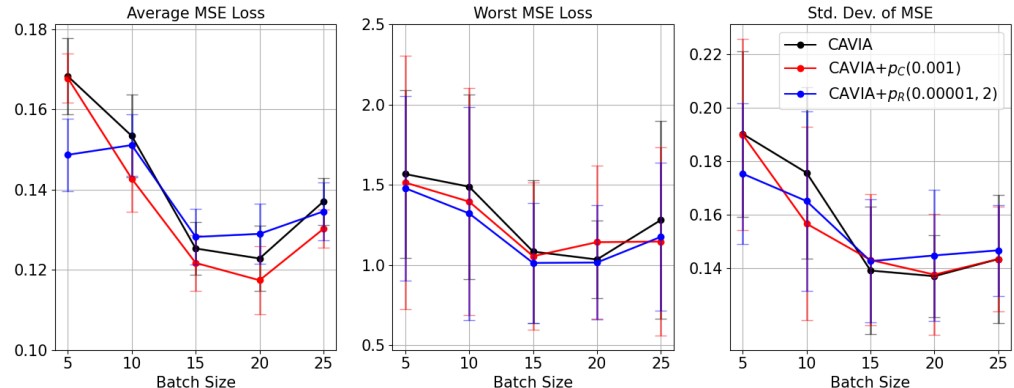

Figure 5: Test MSE statistics with $95\%$ confidence intervals depends on the batch size for 5-shot sinusoid regression. The left plot shows the average MSE loss and the right plot shows the worst MSE loss for CAVIA and the Nash-GBML that combines CAVIA with penalty terms. The statistics are empirical averages over 5000 samples.

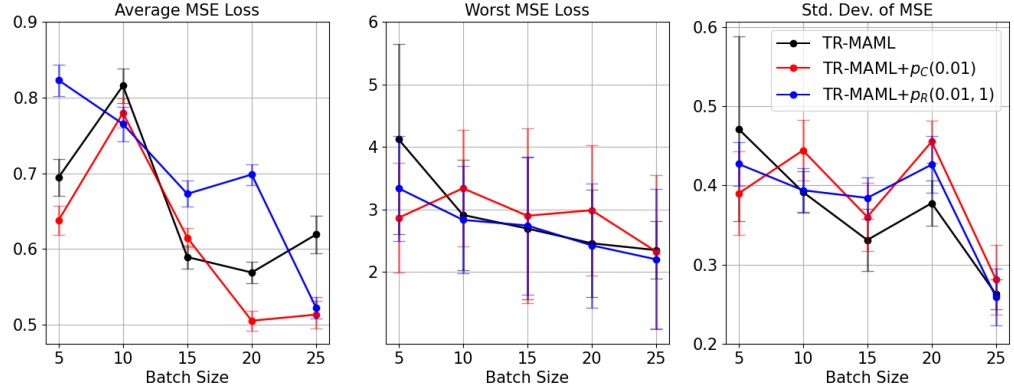

Figure 6: Test MSE statistics with $95\%$ confidence intervals depends on the batch size for 5-shot sinusoid regression. The left plot shows the average MSE loss and the right plot shows the worst MSE loss for TR-MAML and the Nash-GBML that combines TR-MAML with penalty terms. The statistics are empirical averages over 5000 samples.

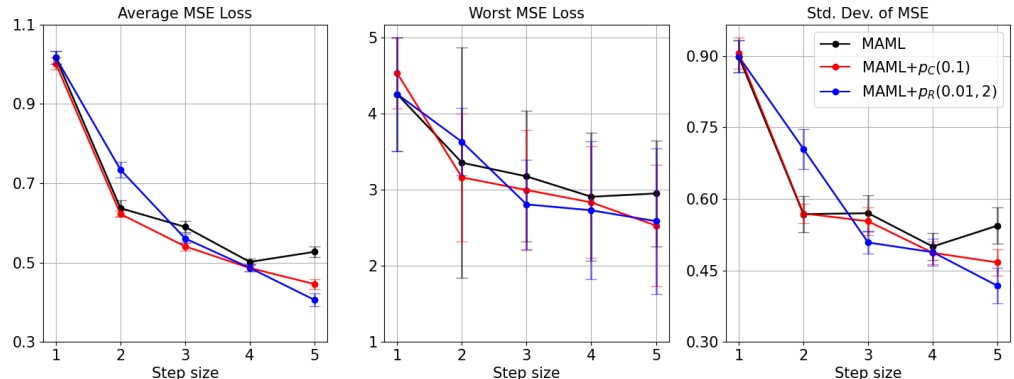

Figure 7: Test MSE statistics with $95\%$ confidence intervals depends on the step size for 5-shot sinusoid regression. The leftmost plot shows the average MSE loss and the center plot shows the worst MSE loss for MAML and the Nash-GBML that combines MAML with penalty terms. The statistics are empirical averages over 5000 samples.

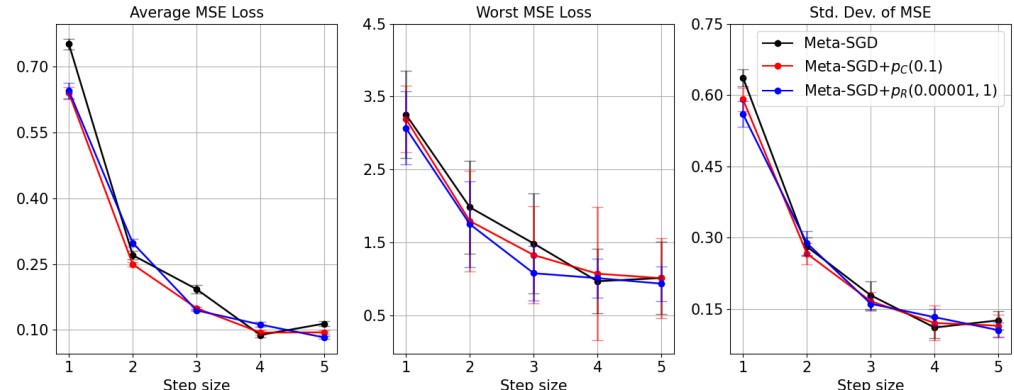

Figure 8: Test MSE statistics with 95% confidence intervals depends on the step size for 5-shot sinusoid regression. The left plot shows the average MSE loss and the right plot shows the worst MSE loss for Meta-SGD and the Nash-GBML that combines Meta-SGD with penalty terms. The statistics are empirical averages over 5000 samples.

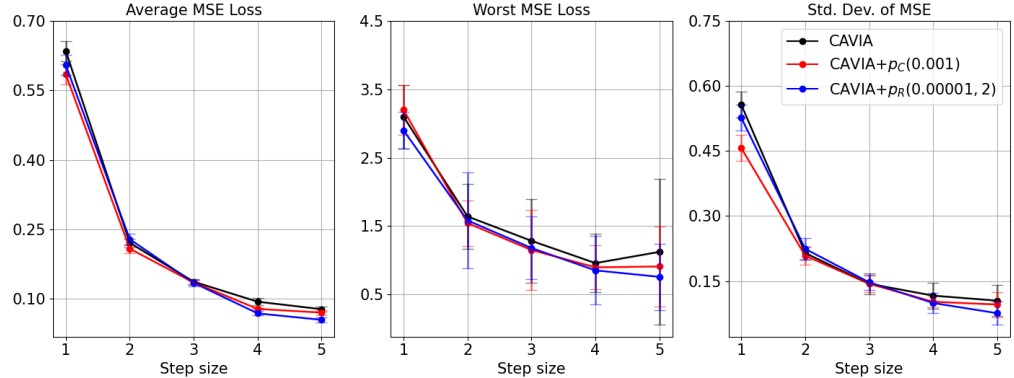

Figure 9: Test MSE statistics with 95% confidence intervals depends on the step size for 5-shot sinusoid regression. The left plot shows the average MSE loss and the right plot shows the worst MSE loss for CAVIA and the Nash-GBML that combines CAVIA with penalty terms. The statistics are empirical averages over 5000 samples.

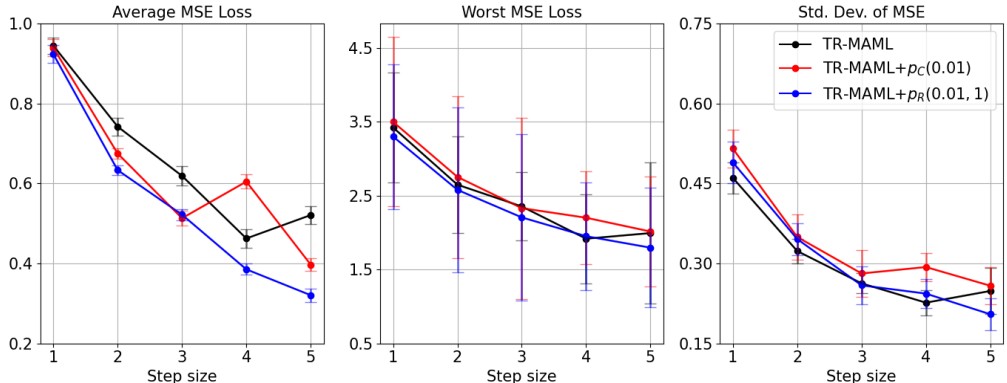

Figure 10: Test MSE statistics with 95% confidence intervals depends on the step size for 5-shot sinusoid regression. The left plot shows the average MSE loss and the right plot shows the worst MSE loss for TR-MAML and the Nash-GBML that combines TR-MAML with penalty terms. The statistics are empirical averages over 5000 samples.

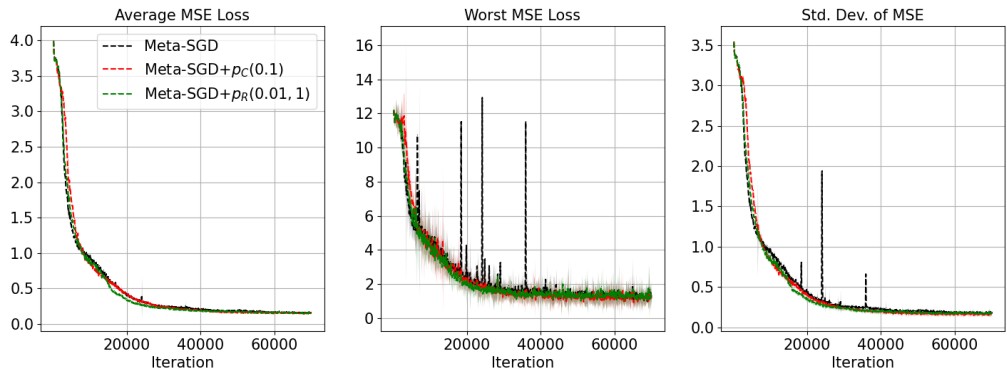

Figure 11: Test MSE statistics with $95\%$ confidence intervals over 5 random trials of Meta-SGD and its Nash-GBML. The leftmost plot shows the average MSE loss, the middle plot shows the worst MSE loss, and the rightmost plot shows the standard deviation. The statistics are empirical averages over 5000 samples.

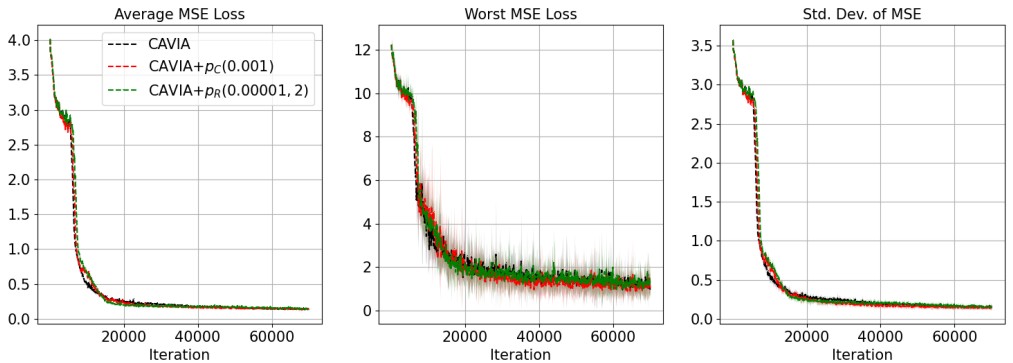

Figure 12: Test MSE statistics with $95\%$ confidence intervals over 5 random trials of CAVIA and its Nash-GBML. The leftmost plot shows the average MSE loss, the middle plot shows the worst MSE loss, and the rightmost plot shows the standard deviation. The statistics are empirical averages over 5000 samples.

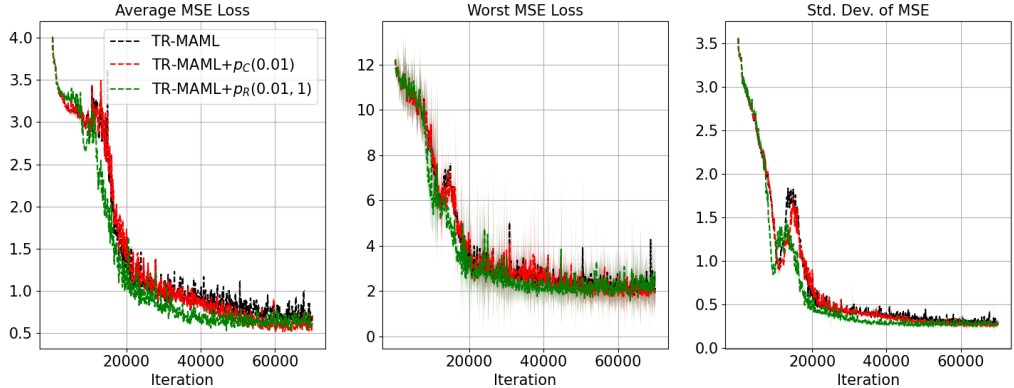

Figure 13: Test MSE statistics with $95\%$ confidence intervals over 5 random trials of TR-MAML and its Nash-GBML. The leftmost plot shows the average MSE loss, the middle plot shows the worst MSE loss, and the rightmost plot shows the standard deviation. The statistics are empirical averages over 5000 samples.

