# OpenReview forum: "Nash-GBML: Nash Gradient-Based Meta-Learning"
_ICLR.cc/2025/Conference — ICLR 2025 Conference Withdrawn Submission_

### Official Review · Reviewer_yoqP · 2024-10-27

**Soundness:** 2
**Presentation:** 3
**Contribution:** 1
**Rating:** 3
**Confidence:** 5

**Summary:**

This paper studies a new scenario of meta-learning that considers the interaction among tasks at the inner level. It formulates meta-learning as a Single-Leader Multi-Follower game and proposes an algorithm, called Nash-GBML to solve the new optimization problem.

**Strengths:**

The paper considers a new formulation of meta-learning.  The setting seems interesting.

The experimental results are complete.

**Weaknesses:**

The motivation for the new meta-learning formulation, which considers the interaction among tasks at the inner level, is unclear and confusing. This paper adds a penalty term in the inner level of MAML, which satisfies the new formulation. This term seems to be related to the robustness of the meta-learning. However, if the paper targets to use the penalty term to address the robustness issue, the paper should be compared with many existing papers that consider the same issue.

The contribution of the paper is weak. The game theoretic interpretation of gradient-based meta-learning (Stackelberg game) is well-known. The proposed method, based on the conventional algorithm for the Stackelberg game, provides limited contribution to the algorithm design.

**Questions:**

The concept of a Single-Leader Multi-Follower (SLMF) game is not clear. As the conventional meta-learning problem has been formulated as a single-leader multi-follower game, what is the difference between the traditional meta-learning formulation and the new proposed formulation?

If the motivation for the penalty term is addressing the robustness issue of meta-learning, the experiment should take the existing robust meta-learning as the baseline.

Confusion about the meta-test stage: if we consider the iteration between different lower-level tasks, how does the method do the meta-test when a single new task is given?

As claimed in the paper, the method with the penalty term holds the benefit of robustness and drawback of the overall average performance. Why the performance of the proposed method can achieve better robustness and better overall average performance than the original version (in Tables 3 and 4)?

---

### Official Review · Reviewer_LL1U · 2024-11-02

**Soundness:** 3
**Presentation:** 2
**Contribution:** 2
**Rating:** 3
**Confidence:** 3

**Summary:**

This paper proposes modeling the optimization procedure of meta-learning as a single-leader multi-follower game, with player interactions encoded by penalty terms that effectively act to constrain the within-task learners. They derive an optimization procedure for their approach and validate it using experiments and theoretical analysis.

**Strengths:**

1. The use of this type of game-theoretic approach to meta-learning is novel and a potential intereseting direction for research, assuming a motivation for tasks not being independent can be found.
2. The authors demonstrate some advantage of their approach experimentally.

**Weaknesses:**

1. The paper states that traditional meta-learning assumes tasks are independent and proposes an approach that should perform better if they are not. However, this setup is not motivated at all: in standard evaluations of meta-learning (few-shot classification, meta-RL, federated learning) the tasks are usually independent, at least in the statistical sense. What is the application here? Within the formulation (e.g. Section 3.1) there is an introduction of penalty terms to “account for the influence of the other task-specific parameters” but what does that actually mean mathematically? Why should the tasks be influencing each other? In Section 3.3 there seems to be some indication of a statistical motivation (“close to each other for effective few adaptations to fit new tasks”) or an optimization motivation (“allowing the algorithm to converge more stably”) but there is no real mathematical justification of either. The exact penalty terms proposed are also not strongly justified beyond allusions to robustness and worst-case vs. average-case.
2. It is unclear to me that the SLMF formulation reflects the goals of meta-learning, which is few sample generalization from unseen tasks. Rather the paper seems to be trying to model the training procedure of meta-learning (without really getting at the statistical aspect).
3. The theory mostly shows correctness but does not suggest any theoretical advantage of using this approach over others.
4. It is unclear to me why the specific penalty terms helped experimentally, or whether similar advantages could not be gained using conceptually simpler regularization approaches. In effect, this issue is similar to the first weakness, in that it is unclear to me what a task interaction is and thus why this approach should be the right way to model it.
5. Code is not provided.

**Questions:**

1. I am not sure if it is accurate to say that “Traditional meta-learning is modeled as the Single-Leader Multi-Follower game” as usually meta-learning is not formulated in game-theoretic terms. Perhaps it is more accurate to say that this paper formulates meta-learning as such a game, shows how past work fits within this formulation, and proposes a generalization of the formulation to non-independent tasks.

---

### Official Review · Reviewer_NC8E · 2024-11-02

**Soundness:** 2
**Presentation:** 2
**Contribution:** 2
**Rating:** 3
**Confidence:** 4

**Summary:**

This paper views meta-learning with a game-theoretic lens, where inner-level learning is considered a Nash game and outer-level learning is considered a Single-Leader Multi-Follower (SLMF) game. Based on these, the authors propose Nash-GBML, which incorporates additional penalty terms (centroid penalty, robust penalty) to improve performance. Convergence analysis is provided.

**Strengths:**

1. Considering game theory for meta-learning is somehow interesting.

**Weaknesses:**

1. The paper is poorly written.
	* The claims are vague. For example, the authors said it considers interactions among tasks compared to previous independent assumptions. However, I do not see an obvious relation between the proposed penalty terms and the task interaction. The penalty terms are very similar to those in previous meta-learning approaches with independent assumptions, such as [1,2, 3]
	* The theorems are badly formulated, with unclear assumptions and undefined notations.
2. Novelty.
	* In terms of the algorithm, I do not find any significant difference from previous works.
3. What is the benefit of using Nash-equilibrium in Algo.2? It's very unclear to me the difference compared to gradient descent.
4. The empirical improvement is very limited.
5. A large amount of related works are missing, which also consider different penalty terms.
      * Probabilistic meta-learning [1, 2, 3].
      * Meta-learning theory with i.i.d. assumption [4,5,6].
      * Meta-learning theory without i.i.d. assumption [7,8,9].
      * Convergence of meta-learning [10, 11].
      * Robust meta-learning [12].


[1] Erin Grant, Chelsea Finn, Sergey Levine, Trevor Darrell, and Thomas Griffiths. Recasting
gradient-based meta-learning as hierarchical bayes. In International Conference on Learning
Representations, 2018.

[2] Jaesik Yoon, Taesup Kim, Ousmane Dia, Sungwoong Kim, Yoshua Bengio, and Sungjin Ahn.
Bayesian model-agnostic meta-learning. In Advances in Neural Information Processing Systems,
pages 7332–7342, 2018.

[3] Chelsea Finn, Kelvin Xu, and Sergey Levine. Probabilistic model-agnostic meta-learning. In
Advances in Neural Information Processing Systems, pages 9516–9527, 2018.

[4] Anastasia Pentina and Christoph Lampert. A pac-bayesian bound for lifelong learning. In
International Conference on Machine Learning, pages 991–999, 2014.

[5] Giulia Denevi, Carlo Ciliberto, Riccardo Grazzi, and Massimiliano Pontil. Learning-to-learn
stochastic gradient descent with biased regularization. In International Conference on Machine
Learning, pages 1566–1575. PMLR, 2019.

[6] Maria-Florina Balcan, Mikhail Khodak, and Ameet Talwalkar. Provable guarantees for gradientbased meta-learning. In International Conference on Machine Learning, pages 424–433. PMLR,
2019.

[7] Anastasia Pentina and Christoph H Lampert. Lifelong learning with non-iid tasks. Advances in
Neural Information Processing Systems, 28, 2015.

[8] Mikhail Khodak, Maria-Florina Balcan, and Ameet Talwalkar. Adaptive gradient-based metalearning methods. arXiv preprint arXiv:1906.02717, 2019

[9] Chen, Qi, et al. "On the stability-plasticity dilemma in continual meta-learning: theory and algorithm." Advances in Neural Information Processing Systems 36 (2024).

[10] Alireza Fallah, Aryan Mokhtari, and Asuman Ozdaglar. On the convergence theory of gradientbased model-agnostic meta-learning algorithms. In International Conference on Artificial Intelligence and Statistics, pages 1082–1092. PMLR, 2020.

[11] Kaiyi Ji, Junjie Yang, and Yingbin Liang. Multi-step model-agnostic meta-learning: Convergence and improved algorithms. arXiv preprint arXiv:2002.07836, 2020.

[12] Wang, Qi, et al. "A simple yet effective strategy to robustify the meta learning paradigm." Advances in Neural Information Processing Systems 36 (2024).

**Questions:**

1. Can the authors elaborate on what kind of interaction you are modelling?
2. Line 208 average or worst-case? Which one exactly?
3. what is the definition of $w(\cdot)$  in eq (13)?
4. In Theorem 3.2 and 3.3, the convex assumption of what?

---

### Official Review · Reviewer_d6AL · 2024-11-04

**Soundness:** 3
**Presentation:** 1
**Contribution:** 2
**Rating:** 3
**Confidence:** 4

**Summary:**

This paper dresses gradient-based meta-learning in a game-theoretic framework with the goal of studying how solving tasks in succession contrasts with solving them independently. In that sense, the authors’ aim to study the interaction among tasks, and the resulting effect on the final meta-learned solution. Based on their framework, they propose two regularization penalties for gradient-based meta-learning. Theoretically, they show convergence and obtain convergence rates in the convex setting. Empirically, they show their penalties outperform MAML on synthetic sinusoid regression tasks, on small image completion tasks, and on image classification tasks.

While not completely novel, the game-theoretic perspective of meta-learning is understudied and this work could help extend it. Unfortunately I found the game-theory language mostly distracting for this work and it’s unclear to me why the paper benefits from it. The proposed penalties can be motivated without it and, as far as I can tell, the convergence results don’t need it either. I also think the convex assumptions for the theory is too strong (see below) and empirically these penalties only marginally outperform the baselines.

**Strengths:**

- I appreciate the authors’ effort to study meta-learning under a different lens — here, game theory. In principle their exposition could help bring meta-learning to the attention of a different community.
- I also appreciated the authors’ choice to include empirical results on non-traditional tasks, namely image completion. Too often, meta-learning papers focus on the same benchmarks (mini-imagenet, cifar-fs) so seeing positive results on new tasks is encouraging. Especially so, since the authors try their proposed method on two meta-learning algorithms and see (minor) gains in both.

**Weaknesses:**

- This paper is difficult to read, in part due to clarity and in part due to the game theory. The clarity issues should be easily fixable: for example, using log-scale for the y-axis of Fig. 2 should help differentiate between methods; similarly, the authors can probably fix their Fig. 2 to highlight the difference between GBML and Nash-GBML — what are the green “interactions among tasks” spring concretely? This point is central to the paper and yet the authors never explicitly define it.

    On the other hand I found the game theory framework unnecessary and ultimately obscures the exposition. I do not understand  what the authors get from it over an optimization or probabilistic framework. The two penalties they introduce are not tied to the framework, nor are the theoretical results. It feels like the authors tried to shoehorn meta-learning into game theory math.
- Theoretically, I think the convexity assumptions is too strong. The gradient-based meta-learning problem is non-convex even for linear regression tasks, so it’s not clear when the paper’s results would apply.
- Empirically, the proposed penalties never significantly outperform the baselines without the penalties. In most results, the baselines get within confidence intervals of the penalized versions despite having one less hyper-parameter. Thus the experimental results alone are not too compelling either.

**Questions:**

Please rebute the weakness claims above. Specifically, could the authors clarify what benefit the game theoretic formulation offers for the proposed penalties?

---

### Note · Authors · 2024-11-13

I have read and agree with the venue's withdrawal policy on behalf of myself and my co-authors.